

**Exploring Diverse Modeling Schemes for Runoff Prediction: An Application to 544 Basins in China**

**Yuqian Hu[a], Heng Li[a] , Chunxiao Zhang[a,b,*] , Dingtao Shen[c,d] , Bingli Xu[e] , Min Chen[f,g,h] , Wenhao Chu[a] , Rongrong Li[i]**

[a] School of Information Engineering, China University of Geosciences in Beijing, Beijing, China.

[b] Observation and Research Station of Beijing Fangshan Comprehensive Exploration, Ministry of Natural Resources, Beijing, China.

[c] Key Laboratory for Geographical Process Analysis & Simulation of Hubei Province, Central China Normal University, 430079, Wuhan, China.

[d] College of Urban and Environmental Sciences, Central China Normal University, 430079, Wuhan, China.

[e] Department of Information and Communication, Academy of  Army Armored Forces, Beijing, China.

[f] Key Laboratory of Virtual Geographic Environment (Ministry of Education of PRC), Nanjing Normal University, Nanjing, Jiangsu, China.

[g] International Research Center of Big Data for Sustainable Development Goals, Beijing, China.

[h] Jiangsu Center for Collaborative Innovation in Geographical Information Resource Development and Application, Nanjing, Jiangsu, China

[i] Institute of Space and Earth Information Science, The Chinese University of Hong Kong, Shatin, New Territories, Hong Kong, China.

*Correspondence to*: C. Zhang (zcx@cugb.edu.cn)

**Key words:** Large-sample hydrological datasets; Deep Learning; LSTM model; Hybrid Modeling; Basin runoff

**Abstract**

Hydrological modeling plays a key role in water resource management and flood forecasting. However, in China with diverse geography and complex climate types, a systematic evaluation of different modeling schemes for large-sample hydrological datasets is still lacking. This study preliminarily constructed a dataset of catchment attributes and meteorology covering 544 basins in China, and systematically evaluated the applicability of process-based models (PBMs), long short-term memory (LSTM) models, and hybrid modeling methods. The results demonstrated: (1) The accuracy of meteorological data critically impacts the prediction performance of hydrological

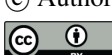



models. High-quality precipitation data enables the model to better simulate the runoff generation
process in the basin, thereby improving prediction accuracy. (2) The hybrid modeling method
possesses regional modeling capabilities comparable to those of LSTM model. It also demonstrates
strong generalization capabilities. In predicting ungauged basins, the hybrid model exhibits greater
stability than the LSTM model. (3) Among the two hybrid modeling methods, the differentiable
hybrid modeling scheme offers a deeper understanding and simulation of hydrological processes,
along with the ability to output unobserved intermediate hydrological variables, compared to the
alternative hybrid modeling schemes. Its prediction results are more consistent with the water
balance of the basin. The research results provide a systematic analysis for evaluating the
applicability of different hydrological modeling methods in 544 basins in China, offering
important guidance for the selection and optimization of future hydrological models.
**1.  Introduction**
1.1 Challenges in hydrological modeling for China's basins
Global water resources management and hydrological process simulation play a key role in
responding to climate change and human activity pressures (Devitt et al., 2023). Accurate
hydrological modeling can provide a basis for scientific decision-making for sustainable water
resources management, prediction of extreme events, and protection of natural ecosystems (Hoy,
2017; Satoh et al., 2022). Among many hydrological variables, daily runoff is particularly widely
used. For example, understanding flow patterns (Brunner et al., 2020), efficient water resources
management (Patle and Sharma, 2023), reservoir operation and flood prediction (Mangukiya and
Sharma, 2025; Mangukiya and Yadav, 2022). However, with the increasing complexity and
uncertainty of basin hydrological processes, the applicability and predictive ability of traditional



process-based models (PBMs) in practical applications face many challenges, such as insufficient
data and complex parameterization.

China is located in a densely populated area in East Asia. The rational management of water

resources is crucial to economic and social development. Therefore, the role of hydrological
modeling in decision support systems has become increasingly prominent (Yin et al., 2018).
However, the diversity of topography and the complexity of climate result in significant variations
in hydrological processes between basins. This has led to the fact that hydrological models based
on physical processes have only achieved certain success in predicting runoff in some basins. For
example, Zhang et al. (2024) used the Xin'an jiang model to predict the runoff in the Suihe and
Zuohe River basins, and X. Liu et al. (2022) integrated the VIC, Xin'an jiang, and DTVGM models
for the Ganjiang River basin. Such models often produce deviations due to insufficient data,
structural design defects, or improper parameterization schemes (Dembélé et al., 2020; Herrera et
al., 2022; Koch et al., 2016; Silvestro et al., 2015), making it difficult to accurately capture the
complex hydrological behavior of the basin. At the same time, with the large-scale growth and
sharing of basin data sets, the development of deep learning technology worldwide has provided
new tools for hydrological modeling. Especially under the condition of sufficient data, deep
learning models such as long short-term memory (LSTM) networks can effectively capture
complex nonlinear relationships and provide high-precision predictions by learning large-sample
hydrological datasets. However, the applicability and predictive ability of deep learning methods
in China's complex hydrological environment have not been fully verified. Hydrological modeling
based on a large-sample hydrological dataset whose basins covering various river systems and
climate regions in China can provide a relatively clear benchmark. This benchmark can assist in



improving relevant hydrological models to solve the runoff prediction problem in China and
catchments with similar basin conditions.
1.2 Hydrological modeling methods
For runoff prediction, existing hydrological modeling methods primarily include process-
based models, deep learning models , and hybrid modeling methods that couple the two. Process-
based models focus on the physical mechanisms of hydrological processes, provide theoretical
support and interpretability, and are particularly suitable for explicitly describing the dynamic
hydrological behavior of a basin. However, this type of model emphasizes the uniqueness of
hydrological processes and shows that it is impossible to effectively solve different hydrological
processes in different basins with a unified approach (Blöschl et al., 2019). Even when used to
model hydrology for a single basin, such models typically require a substantial amount of high-
quality input data and involve a degree of subjectivity and complexity in the parameterization
process. This significantly impacts their applicability and accuracy.
In contrast, deep learning models, such as long short-term memory networks (LSTM), learn
the dynamic characteristics of basin hydrological processes through historical data, can effectively
capture nonlinear relationships, and do not need to rely on precise physical parameters as PBMs.
LSTM has attracted great interest in the field of hydrology, and has shown high prediction accuracy
and stability in predicting various hydrological variables, including soil moisture (Fang et al., 2017;
J. Liu et al., 2022; O. and Orth, 2021), runoff (Feng et al., 2020; Kratzert et al., 2019a), river
temperature (Qiu et al., 2021; Rahmani et al., 2023), and dissolved oxygen (Kim et al., 2021; Zhi
et al., 2021). This type of model excels in data collaboration (Fang et al., 2022) and therefore
thrives in big data environments (Kratzert et al., 2019a; Tsai et al., 2021). This type of model
performs well in data-driven collaboration and thrives in big data environments. However, deep



learning (DL) methods are not without limitations. On one hand, they may struggle with poor
extrapolation ability in the absence of physical constraints, which can limit the model's
generalization capability when confronted with unknown climate conditions or long-term
predictions, potentially resulting in increased prediction errors and even physically unreasonable
results. On the other hand, compared to PBMs that offer clear explanations of physical mechanisms,
the predictive results of DL lack interpretability. The variable relationships represented by DL may
rely on statistical correlations present in the data rather than true physical causal relationships,
which affects its applicability under varying hydrological conditions.

To address the limitations of the two aforementioned modeling approaches, hybrid modeling

methods have been proposed (Konapala et al., 2020; Willard et al., 2021). The integration of deep
learning models and process-based models (PBMs), along with the development of hybrid systems,
has been recognized as a promising method for effectively enhancing runoff prediction (Slater et
al., 2023). This modeling scheme not only retains the theoretical foundation and interpretability of
PBMs but also leverages the robust data fitting capabilities of DL. Common hybrid schemes
currently include alternative hybrid modeling scheme and differentiable hybrid modeling scheme.

Alternative hybrid modeling scheme commonly employs DL as a post-processing tool to

adjust the discrepancies between PBM outputs and actual observations by utilizing PBM outputs
as additional input features. For instance, runoff, soil moisture, and snow cover are used as inputs
to DL (Amendola et al., 2020; Frame et al., 2021; Wang et al., 2023). Such schemes have been
tested in various regions (Cho and Kim, 2022; Shen et al., 2022). DL is trained in a supervised
manner, involving only minor modifications to the existing PBMs, which allows for the
preservation of valuable physical knowledge inherent in the PBM.





Differentiable hybrid modeling scheme is made possible by the emergence of automatic
differentiation (auto-diff) technology (Baydin et al., 2018). Automatic differentiation technology
calculates gradients automatically through the chain rule, thus avoiding the trouble of manually
deriving derivatives and expediting the computation of complex gradients in deep neural networks.
This technological advancement addresses the challenge of combining DL with PBMs and
significantly enhances the integration of deep learning models into process-based models.
Specifically, this hybrid modeling approach neuralizes the process-based model and adjusts model
parameters by back propagating gradients based on daily prediction results. This method retains
the interpretability of the process model while improving prediction accuracy through deep
learning. Additionally, within this unified differentiable architecture, neural networks can
selectively replace inaccurate process representations and parameterizations found in process-
based models. Due to the innovative nature of this approach, it has garnered significant attention
from researchers and has led to the development of multiple variants and adaptations (Frame et al.,
2021; Höge et al., 2022; Jiang et al., 2020; Li et al., 2023).
1.3 Research gap and study objectives
There remains a gap in research regarding the applicability and performance comparison of
various hydrological modeling methods in China basins. This highlights a notable research deficit
in the field of hydrological modeling for large-sample hydrological datasets of China. Although
some scholars have explored different modeling approaches and conducted theoretical analyses
and empirical studies, the applicability and predictive capabilities of existing models still require
systematic evaluation, especially in the context of China's complex hydrological environment.
Specifically, several issues persist in the current research: (1) Comparative studies primarily focus
on specific river basins or small sample datasets. For instance, the studies conducted by Wu et al.



(2024), Dong et al. (2024), and Xu et al. (2023) centered on small-scale river basins, namely the
Weihe River, Lancang River, and Yellow River, respectively. The lack of systematic evaluation
across large-sample hydrological datasets limits the feasibility of model promotion and application.
Therefore, a systematic evaluation based on large-sample hydrological datasets can not only
enhance the generalization ability of the model but also provide a more reliable reference for
hydrological predictions nationwide. (2) The guiding principles for model selection in basins with
different climatic and attribute characteristics are not yet clear, leading to a lack of relevance in
practical applications and difficulty in identifying the optimal modeling methods. In-depth
research is needed on the adaptability of different hydrological models in various basin
environments, which will help improve modeling efficiency and enhance the stability of
predictions. (3) Existing research on the application effects of hybrid modeling methods in China
basins, along with their comparative analysis with single models, remains relatively insufficient,
resulting in a lack of comprehensive understanding of the relative advantages of different hybrid
modeling methods. This limits the further development and optimization of hybrid modeling
methods in hydrological prediction. A systematic evaluation of the performance of hybrid
modeling methods can not only reveal their potential advantages over traditional methods but also
provide directions for future model improvements, offering more accurate and reliable tools for
simulating and predicting complex hydrological systems.

To address these issues—namely, the limited large-sample studies, unclear model selection

guidelines, and insufficient analysis of hybrid models—this study systematically evaluates the
applicability of three common hydrological modeling approaches using a comprehensive large-
sample hydrological dataset for China. By comparing and analyzing the performance of different
models in daily runoff prediction, the study seeks to provide scientific guidance for the selection



and application of hydrological models. Specifically, the dataset encompasses 544 basins across
nine river systems and seven climate regions in China. It integrates multiple data sources, including
remote sensing products and reanalysis data, to relatively accurately describe basin attributes and
meteorological characteristics. For each basin, the dataset includes vector boundaries and time
series data. The static attribute dataset consists of 6 categories and 15 types. Hydrological and
climate attributes are derived by calculating relevant indices from the aforementioned
meteorological data, while topographical, soil, vegetation, and geological characteristics are
extracted from publicly available global datasets. The hydrological models evaluated in this study
include: process-based models (EXP-HYDRO model, Xin'an jiang model), deep learning models
(LSTM), alternative hybrid hydrological models (EXP-IN-LSTM, XAJ-IN-LSTM), and
differentiable hybrid hydrological models (EXP-dPL, XAJ-dPL). To ensure a fair evaluation, the
training periods, testing periods, and the prediction in ungauged basins (PUB, Hrachowitz et al.,
2013; Sivapalan et al., 2003) scheme for all models were consistent. In addition to prediction
performance, this study also assessed the water balance of each basin using the prediction results
of different models. Based on this work, the study not only reveals the applicability of different
models in China's basins but also fills the gap in existing research on model comparison in large-
sample hydrological datasets and complex basin environments. Furthermore, it provides a
performance benchmark and guidance for the selection and improvement of hydrological modeling
methods in China in the future.



**2. Data**
2.1 Study Areas

In China, obtaining a datasets for Large-scale hydrological studies is challenging for two

main reasons. Firstly, accurate daily runoff observation data often needs to be kept confidential.
Secondly, the start and end times, as well as the quality of data from hydrological observation
stations across different regions, can vary significantly. Additionally, the diverse topography and
climate conditions in China mean that some watersheds lack essential meteorological and
hydrological observation stations (Meng et al., 2017). The primary goal of this study is not to
provide a reliable hydrological dataset. Instead, the basic requirement for the dataset is to ensure
that the catchment areas encompass a range of climatic and topographic conditions found across
China. To meet these requirements, global river network data (Lehner et al., 2008) and DEM
elevation data were utilized to delineate the basin vector boundaries. The outlet locations of each
catchment (see Figure S1 in Supplementary Materials) were determined using the D8 flow
direction scheme. Furthermore, to facilitate the extraction of meteorological forcing data, basins
smaller than 1000 km² were excluded from the analysis. Ultimately, the vector boundaries of 544
basins were delineated, as shown in Figure 1(a). These 544 basins represent a variety of terrains—
including plateaus, plains, and mountains—with average altitudes ranging from 0 to 5000 m. The
catchment areas included in this dataset span various types of basins within China's nine river
systems (Figure 1(c)) and seven climate regions (Figure 1(d)). Detailed information regarding each
river system and climate zone is presented in Table S1 and Table S2 of Supplementary Materials.




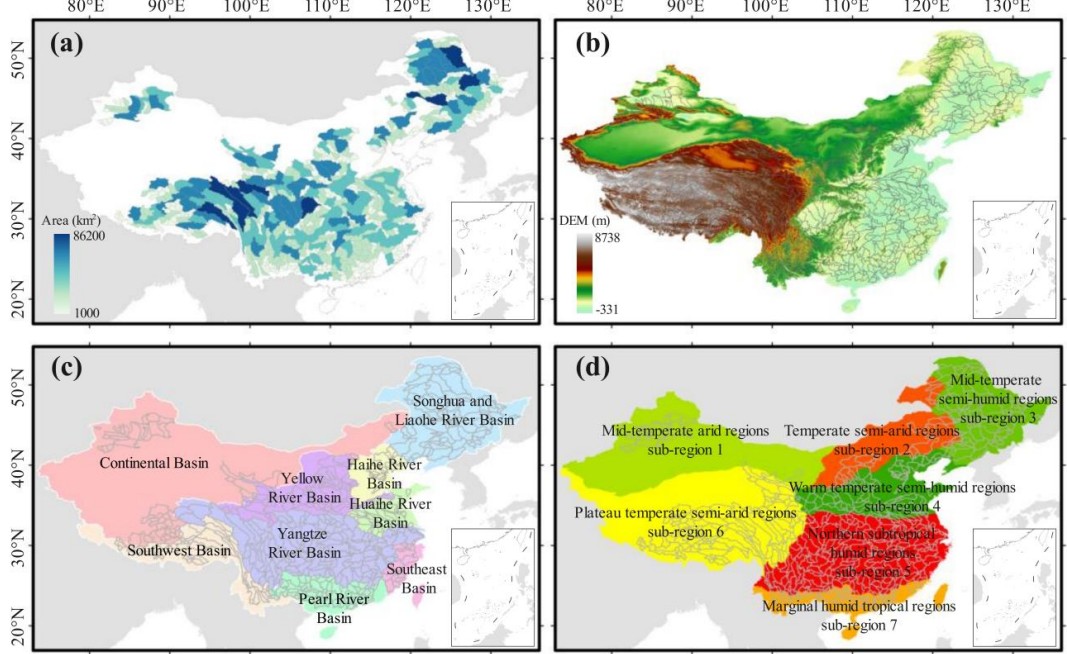


**Figure 1. Spatial distribution of 544 basins.** Basin boundaries (a) and areas (b) of the 544

basins included in this study. The six defined macro-zones are indicated in blue and purple

arrows. (b) and (d) are the divisions of China's nine river systems and seven climate regions,

respectively. (The map of China used in this study is from https://www.tianditu.gov.cn/.)

2.2 Meteorological forcing and Runoff

The meteorological data used in this study are sourced from the ERA5 (Hersbach et al.,

2020) and CN05.1 datasets (Gao et al., 2013). The meteorological forcing elements and their units

provided by these datasets are shown in Table 1. We selected these two datasets to extract basin-

scale meteorological elements based on the following considerations: (1) ERA5 dataset: Although

previous studies (Jiao et al., 2021; Liu et al., 2024) have demonstrated that the meteorological data

from ERA5 exhibit certain deviations in Asia, the ERA5 dataset still offers significant advantages.



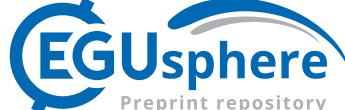

It not only provides a wide variety of meteorological element types but also has an extensive daily
meteorological data time span, ensuring dataset completeness. (2) CN05.1 dataset: This dataset is
interpolated from observational data collected from over 2,400 meteorological stations across
China. Therefore, it can relatively accurately characterize the trends in meteorological changes in
the country and offers high spatial resolution and applicability. Utilizing global-scale
meteorological data (such as ERA5) alongside more precise meteorological data for specific
research areas (such as CN05.1) to evaluate the applicability of different hydrological models can
not only enhance the robustness of the evaluation results but also help to verify the differences and
applicability of various meteorological data in hydrological predictions.
**Table 1** *Meteorological forcings data for 544 basins.*

| Variable name | Description | Unit |
|---|---|---|
| total_precipitation_sum | Average daily precipitation | m |
| temperature_2m | 2 m daily mean air temperature | K |
| potential_evaporation_sum | Total potential evapotranspiration | m |
| surface_pressure | Near-surface daily average | Pa |
| surface_solar_radiation_downwards_sum | Short-wave radiation | $J/m^2$ |

The runoff data provided by the originates from VIC-CN05.1 dataset (Miao and Wang, 2020),
which is consistent with the total runoff simulated by the Global Runoff Data Center
(UNH/GRDC). Due to the high confidentiality surrounding China's runoff observation data, the
number of basins with available runoff data is limited, and the start and end times of the runoff
time series vary among different basins. Relying solely on available observational data for
calibration may lead to systematic deviations in the runoff data across all basins. The daily runoff
time series of each basin outlet were obtained. In order to verify the feasibility of this method, the
daily runoff time series of 15 basin outlets with runoff observation data and the runoff hydrograph

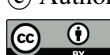



of actual observation data are shown in Supplementary Figure S2. It can be seen that although it
has not been strictly calibrated, the daily runoff time series extracted using the above method can
basically reflect the actual runoff change trend of the basin. The aim of this study is not to provide
a highly precise basin hydrological dataset but to construct a relatively comprehensive large-
sample dataset encompassing meteorological, hydrological, and attribute data for basins across
different topographic and climatic regions of China.

2.3 Static catchment attributes

When performing regional modeling, the static attributes of the basin are used as inputs to

the model, facilitating the model's ability to learn and extract information related to rainfall-runoff
behavior while distinguishing between different basins. This enables the model to interpret the
unique characteristics of each basin, thereby improving the accuracy of runoff predictions, as
demonstrated by Kratzert et al. (2019b). Therefore, in addition to extracting daily meteorological
forcing and runoff data for each basin, the dataset used in this study also includes static basin
attribute data. This dataset comprises 15 attributes categorized into six groups: meteorology,
hydrology, topography, soil, vegetation, and geology. Among these, meteorological and
hydrological attributes are calculated based on the meteorological and runoff time series of each
basin. Other static attribute data are derived from global data products based on the vector
boundaries of each basin. The abbreviations, meanings, and sources of each static attribute are
presented in Table 2. While existing large-sample hydrological datasets (such as CAMELS (Addor
et al., 2017) and Caravan (Kratzert et al., 2023)) offer a richer variety of static attributes, our
research goal is not to pursue dataset completeness. Instead, we aim to utilize relatively available
datasets to evaluate the performance of different models in China. The six categories of static
attributes broadly cover the primary factors affecting the hydrological behavior of the basin.





Among them, meteorological and hydrological attributes have the most direct and significant
impact on runoff, while topography, soil, vegetation, and other attributes can help model to
enhance the understanding of basin characteristics from the perspectives of spatial distribution and
hydrological processes. Additionally, these static attributes can be obtained through global data
products and basin vector boundary calculations, reducing reliance on region-specific data.
**Table 2** *Static basin attributes data for 544 basins.*

| Variable name | Description | Unit | Source |
|---|---|---|---|
| area | Basin area | km$^2$ | This study |
| srftopo | Surface (rock + ice) elevation | m | Amante and Eakins (2009) |
| slope_avg | Mean subgrid slope (inner slope) | m/m | Amante and Eakins (2009) |
| wcap | Maximum soil water capacity | Kg/m$^2$ | Hagemann and Stacke (2015) |
| wava | Plant available water | Kg/m$^2$ | Hagemann and Stacke (2015) |
| fveg | Fractional vegetation cover climatology relative to LSM | / | Hagemann (2002) |
| lai | Leaf area index | m$^2$/m$^2$ | Hagemann (2002) |
| p_mean | Mean daily precipitation | m | This study |
| pet_mean | Mean daily potential evapotranspiration | m | This study |
| aridity | Ratio of Mean PET to Mean Precipitation | - | This study |
| frac_snow | Fraction of precipitation falling on days with temp < 0 ∘C | - | This study |
| high_prec_freq | Frequency of days with ≤ 5× mean daily precipitation | - | This study |
| high_prec_dur | Average duration of high precipitation events | - | This study |
| low_prec_freq | (number of consecutive days with ≤ 5× mean daily precipitation) Frequency of dry days (< 1 mm/day) | - | This study |
| low_prec_dur | Average duration of dry periods | - | This study |
| | (number of consecutive days with precipitation < 1 mm/day) | | |

To facilitate reference and application by researchers in other regions, we have made the
global data products used and the code for extracting watershed attributes publicly available. We
hope this will enhance the transparency and reproducibility of our research, making it easier for
researchers in other regions to replicate and expand upon our methods.
**3. Methodology**
3.1 Experimental design
When extracting data based on the basin vector boundary, it is essential to ensure the
consistency of the time span for both runoff and meteorological data. To eliminate potential





deviations caused by differing start and end times across various data sources, the time range for
all data is standardized from October 1, 1975, to September 30, 2015. For different hydrological
models, the training and testing periods are uniformly established: the training period spans from
October 1, 1975, to September 30, 1995, while the testing period extends from October 1, 1995,
to September 30, 2015. This consistent division facilitates an accurate evaluation of the fitting and
predictive capabilities of different models concerning the runoff process. Furthermore, to further
assess the generalization performance of the models, a five-fold cross-validation method is
implemented. Specifically, the 544 basins are divided into five relatively even clusters (with each
cluster containing either 109 or 108 basins, as shown in Figure 2). The validation process is as
follows: the model is trained using the training period data from the basins in nine of the clusters,
and its performance is validated on the test period data from the remaining cluster. This operation
is repeated in a loop, with each iteration designating a different cluster as the ungauged basin,
thereby allowing for the evaluation of the predictive performance of each basin treated as an
ungauged basin.





*Step 1 Data preprocessing*

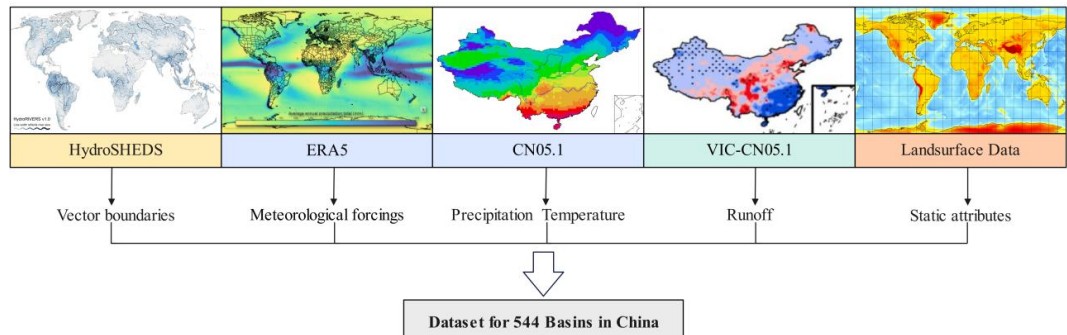

*Step 2 Model Performance Evaluation*

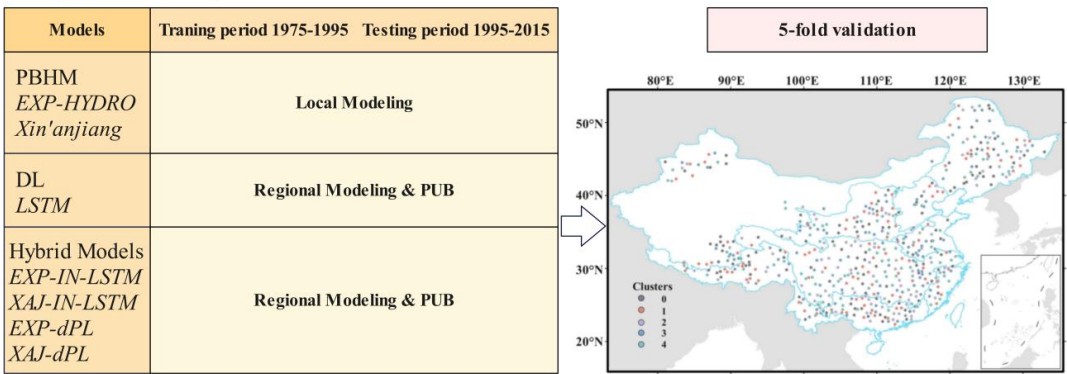

*Step 3 Evaluation of water budget closure*

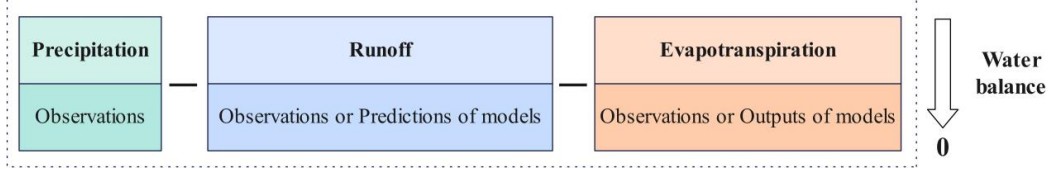


**Figure 2. Overview of the experimental design.**
3.2 Process-based models
The PBMs evaluated in this study include EXP-HYDRO model (hereafter called EXP) (Patil
and Stieglitz, 2014) and the Xin'anjiang model (hereafter called XAJ) (Ren-Jun, 1992). The model
architectures of both are shown in Figure 3. EXP-HYDRO is a conceptual hydrological model that
operates on a daily time step and adheres to the law of water balance. The inputs driving the model
consist of precipitation, temperature, and day length. The hydrological variables that the model



can output include snow accumulation, snowmelt, evapotranspiration, soil moisture, and runoff.
The calibration of the model is governed by six physically meaningful hydrological parameters
(the specific meanings of these parameters are provided in Table S3 of Supplementary Materials).
The original version of the Xin'anjiang model is a rainfall-runoff model designed for hydrological
forecasting in humid and semi-humid regions (Ren-Jun, 1992). To enhance the model's
applicability under complex climatic conditions, this study employs a simplified version of the
Xin'anjiang model, as depicted in Figure 3(b). The simplified model retains four main subprocesses
from the original Xin'anjiang model: evapotranspiration, runoff generation, runoff separation, and
runoff routing. While the original model considered the impact of impervious surfaces on runoff,
this aspect has been simplified in the current study to better align with research needs and data
characteristics. Nevertheless, the model retains its core hydrological processes and can effectively
simulate the hydrological behavior of the basin. The evapotranspiration module accounts for
evaporation from three layers of soil (upper, lower, and deep layers) and calculates the
evapotranspiration for each layer using an empirically defined formula based on the soil moisture
content and evapotranspiration rates. The runoff generation module simulates moisture
distribution and initiates runoff formation. The generated runoffs are subsequently separated into
surface runoff, convergence, and groundwater by the runoff separation module based on free water
storage. Finally, the runoff routing module directs the water flow to the basin outlet for surface
runoff and into linear reservoirs for groundwater flow. A total of 8 hydrological parameters are
integrated into the adjusted Xin'anjiang model. Further details can be found in Table S4 of
Supplementary Materials.





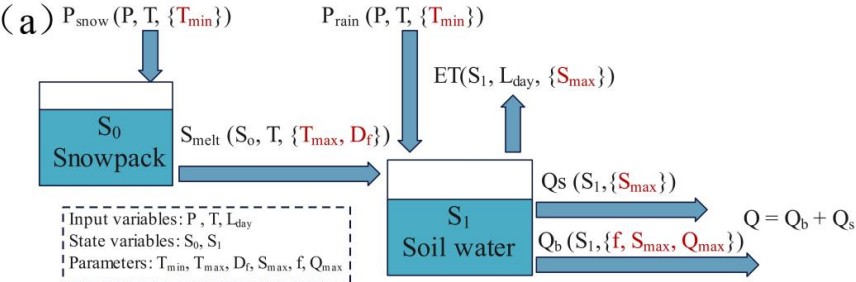

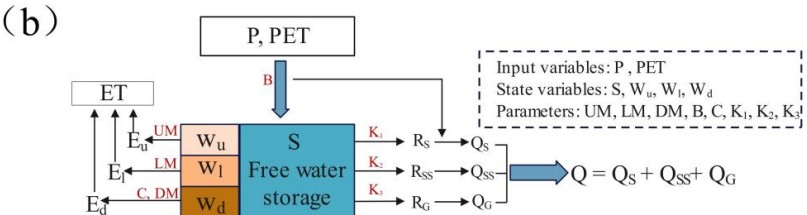


**Figure 3. The structure of the EXP-HYDRO (a) and adapted Xin'anjiang model (b).**


3.3 Deep learning model

With the continuous advancement of deep learning technology, its applications in the field of

hydrology are also expanding. This study utilizes the classic LSTM model as a representation of a
purely data-driven approach. LSTM, a type of recurrent neural network (RNN), was first proposed
by Hochreiter and Schmidhuber (1997). Kratzert et al. (2018) applied the LSTM model to rainfall-
runoff modeling. The specific structure of the LSTM model can be described by the following
equations:
$$i[t] = \sigma(W_i x[t] + U_i h[t-1] + b_i) \tag{1}$$

$$f[t] = \sigma(W_f x[t] + U_f h[t-1] + b_f) \tag{2}$$

$$g[t] = tanh(W_g x[t] + U_g h[t-1] + b_g) \tag{3}$$

$$o[t] = \sigma(W_o x[t] + U_o h[t-1] + b_o) \tag{4}$$

$$c[t] = f[t] \odot c[t-1] + i[t] \odot g[t] \tag{5}$$

$$h[t] = o[t] \odot tanh(c[t]) \tag{6}$$





where i[t], f[t], and o[t] are the input, forget, and output gates, respectively, g[t] is the cell input,
x[t] is the network input at time step t ($1 \leqslant t \leqslant T$), and h[t-1] is the recurrent input c[t-1] of the
cell state at the previous time step. $\sigma(\cdot)$ is the sigmoid activation function, $\tanh(\cdot)$ is the hyperbolic
tangent function, and $\odot$ is an element-wise multiplication. Intuitively, the cell state (c[t])
characterizes the memory of the system. These are modified by a combination of (i) the forget gate
(f[t]) and (ii) the input gate (i[t]) and the cell update (g[t]). Ultimately, the output gate (o[t])
controls the flow of information from the state to the model output.
The LSTM model architecture used in this study to predict daily runoff in China consists of
a hidden layer with 256 hidden units. The regional LSTM model uses 20 input features: 5
meteorological forcings and 15 static basin attributes. All input data are normalized before training.
Additionally, to ensure that the model learns the temporal variation of runoff while taking into
account training efficiency, we input a sequence of 365 days into the network for each batch size
and use a sliding window of 31 days to learn all sequences. Furthermore, to understand the impact
of basin attributes on runoff, the operation of basin attributes across different basins also follows
the aforementioned sliding window learning rules. The LSTM employs the Adaptive Moment
Estimation algorithm (Adam, Kingma and Ba, 2014) to estimate model parameters. The initial
learning rate is set to 0.01, and the maximum number of training iterations is set to 150. All the
LSTM models are trained using the specified hyperparameters.
3.4 The hybrid models
Both alternative hybrid modeling and differentiable hybrid modeling schemes are designed
to couple the advantages of PBM and DL models. The architectures of the two types of models are
shown in Figure 4. PBM represents both the EXP and XAJ models, which are utilized to extract
and characterize the physical mechanisms involved in hydrological processes. LSTM serves as





either a post-processing tool or a differentiable component, integrating with PBM to create
alternative hybrid models and differentiable hybrid models. In the alternative hybrid modeling
scheme depicted in Figure 4(a), LSTM functions as a post-processing tool to adjust discrepancies
between the PBM outputs and observations. This model's input includes not only the results
predicted by PBM but also the input variables of the pure data-driven model, including
meteorological forcing and static basin attributes. This approach allows the alternative hybrid
modeling method to retain the explanatory power of the physical mechanisms inherent in the
process model while leveraging LSTM's capability to effectively capture nonlinear relationships,
thereby compensating for the limitations of PBMs in large-sample hydrological datasets and
complex basins. In the differentiable hybrid modeling scheme shown in Figure 4(b), discrete
ordinary differential equations based on PBM are encoded into standard recurrent neural network
(RNN) units, encompassing the water balance law and fundamental hydrological processes.
Simultaneously, the neural network acts as a parameterized channel, incorporating static attributes
as additional input variables into the overall model framework for joint optimization. This enables
the model to dynamically adjust hydrological parameters based on the characteristics of basin
attributes, overcoming the traditional physical process models' reliance on fixed parameterization,
and adapting to runoff relationships across different basins and climatic conditions.





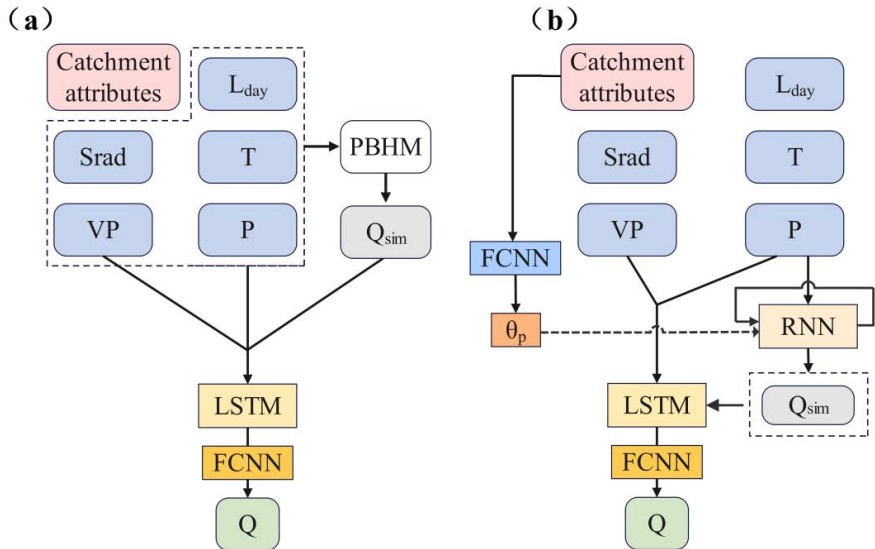


**Figure 4. The structure of the hybrid hydrological models.**

Specifically, the hybrid hydrological models developed in this study include four types: (1) a
alternative hybrid model that uses EXP-predicted runoff values as an additional input to LSTM
(EXP-IN-LSTM); (2) a alternative hybrid model that uses XAJ-predicted runoff values as an
additional input to LSTM (XAJ-IN-LSTM); (3) a differentiable hybrid model that retains the EXP
structure while incorporating the LSTM network for parameter learning (EXP-dPL); and (4) a
differentiable hybrid model that retains the XAJ structure while incorporating the LSTM network
for parameter learning (XAJ-dPL). The details for all the hybrid models trained and tested in this
study—including input data, training sets, testing sets, and inputs—are presented in Table S5 of
Supplementary Materials.
**4. Results and discussion**
4.1 Meteorological forcing assessment
Figure 5 shows the spatial distribution of average annual precipitation and average daily
temperature for each station under different datasets. Overall, although neither of the two





meteorological datasets has been corrected using station observation data, their spatial
distributions are largely similar. Precipitation exhibits a decreasing trend from the southeastern
coast to the northwestern inland areas. Temperature decreases with increasing latitude in the east,
and the temperatures in the Qinghai-Tibet Plateau are relatively low due to the influence of altitude
and terrain. It should be noted that by observing the frequency distribution curves of the average
annual temperature and average daily temperature of 544 basins, it can be seen that the overall
temperature distribution in the two datasets is quite similar. However, the distribution of the
precipitation data shows significant differences. The precipitation data provided by ERA5 varies
greatly between different basins, showing some extremely wet or extremely dry basins, while the
precipitation data provided by CN05.1 is relatively uniform across basins.



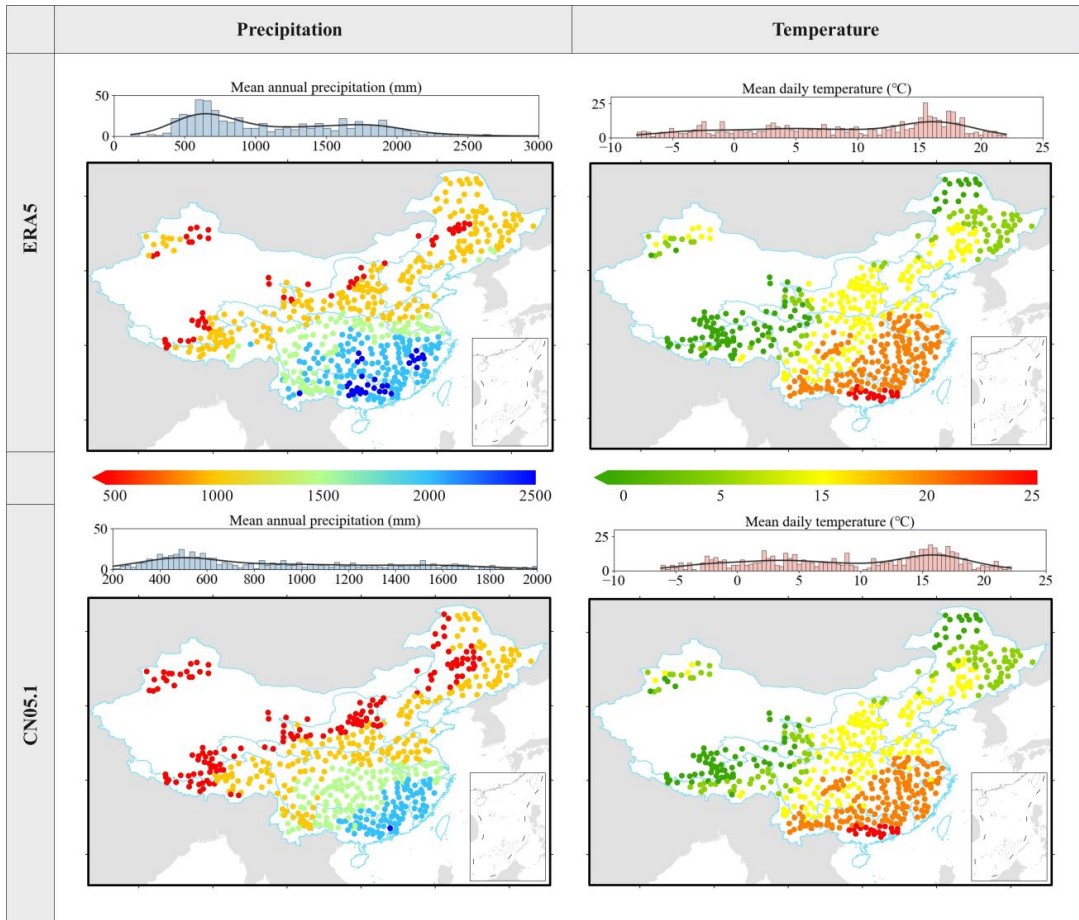

**Figure 5. Spatial distribution of precipitation and temperature in 544 basins using ERA5 and CN05.1 Datasets.**

In order to further explore the differences and systematic biases in various meteorological datasets, the Budyko framework (Budyko and Miller, 2010) was employed to determine the impact of different precipitation data on the water balance of various basins. This framework links climate with basin runoff and evapotranspiration in a simple and intuitive manner, aiming to facilitate the analysis of how evapotranspiration and runoff in each basin are influenced by available energy and precipitation. Figure 6 shows a scatter plot of the evaporation index (EI, the ratio of annual average evapotranspiration to annual average precipitation) and the drought index. When using the same runoff and evapotranspiration data, the precipitation data provided by ERA5 resulted in more





basins (111) violating the water-heat balance. In these relatively humid basins, the relationship
between precipitation and evaporation did not adequately satisfy the water-heat balance conditions.
In contrast, when using CN05.1 data, fewer basins (12), concentrated in humid basins at medium
and high altitudes) violated the water-heat balance. It should be noted that this difference may be
related to the quality and processing methods of the data source. On one hand, ERA5 is a global
meteorological model, and its precipitation data has not been corrected using station observation
data, which may lead to significant deviations. On the other hand, the runoff data product used in
this study was simulated by the VIC model, which uses CN05.1 meteorological data. VIC (Liang
et al., 1994) is a hydrological model based on physical processes, incorporating the equations of
water conservation and energy conservation into its implementation logic to ensure the scientific
and physical consistency of the simulation. This results in fewer basins violating the water-heat
balance when using CN05.1 precipitation data to validate the Budyko framework. When
establishing large-sample hydrological datasets cross different regions, it is necessary to consider
the water budget balance at the watershed scale. We recommend that future studies aimed at
creating accurate large-sample hydrological datasets in China should consider the water and
energy balance of each watershed while fully calibrating each dataset, especially for high-altitude
watersheds that are relatively humid.





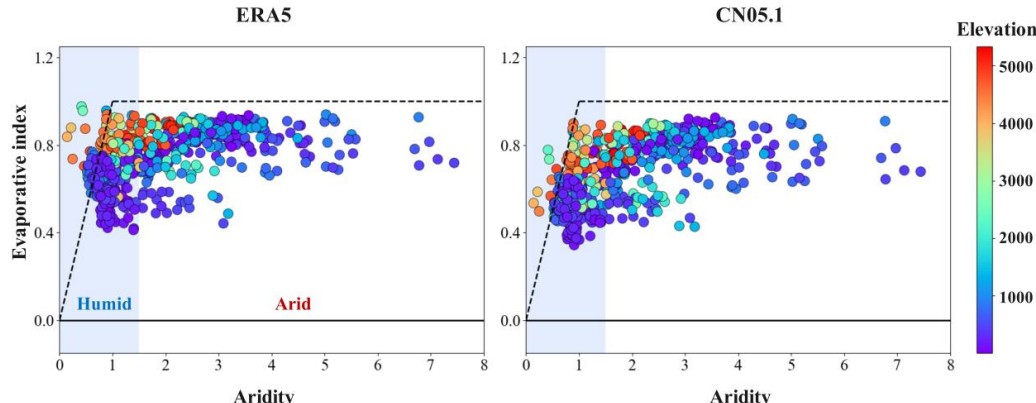

**Figure 6. Water balance for 544 basins, illustrated in a Budyko scheme for ERA5 (a) and CN05.1 (b). Markers are coloured by the basin mean elevation.**

4.2 Performance comparison of process-based models

Figure 7 shows the runoff prediction performance of different PBMs for 544 basins. Overall,

both the EXP and XAJ models demonstrate greater accuracy in representing the rainfall-runoff

relationship for the wetter basins in the southeast compared to the inland basins in the northwest.

The prediction skills of both PBMs show significant improvement when using precipitation data

from CN05.1 as input, in contrast to using ERA5 precipitation data. Notably, for the XAJ model,

the median NSE across the 544 basins reaches 0.63 when using CN05.1 precipitation data.





**Figure 7. Performance of process-based hydrological models during the testing period (1995.10.1–2015.9.30). Spatial distributions of the Nash-Sutcliffe efficiency (NSE) for EXP-HYDRO model and Xin'anjiang model. The NSE colormap is capped within [0, 1] for better visualization.**

Furthermore, the specific performance differences of different PBMs in each basin are

compared. Figures 8(a) and 8(b) show which PBMs exhibit better prediction performance in each



basin under varying precipitation data. In general, the source of precipitation data significantly
influences the selection of PBMs for most basins. This means that for the same basin, if the input
precipitation data sources differ, the choice of PBM should be adjusted accordingly. Additionally,
some basins show consistent selection preferences under the two sets of precipitation data. For
instance, in most basins located in the middle and lower reaches of the Yangtze River and the
southeastern rivers, which are relatively humid, the XAJ model is a better choice than the EXP
model under different precipitation data. Despite the differences in precipitation data, the EXP
model consistently outperforms the other models for most basins in the Haihe River system. For
nearly half of the basins (247), the EXP model demonstrates superior prediction performance when
using ERA5 precipitation data, while the XAJ model performs better with CN05.1 data. This
indicates that the applicable PBM for the same basin may change solely due to variations in
precipitation data. Thus, in addition to the climatic characteristics of the basin, the accuracy and
reliability of meteorological data play a crucial role in determining the prediction performance of
PBMs.



Figure 8. Comparison of EXP-HYDRO model and Xin'anjiang model performances in different basins using ERA5 and CN05.1 precipitation data.



4.3 Performance and generalization of LSTM
Figure 9 shows the prediction performance of the purely data-driven LSTM model in regional
modeling and PUB under different precipitation data sources. Similar to performances in large-
sample hydrological datasets of other regions, the LSTM model also demonstrates strong
capabilities in handling large amounts of hydrological data and excels in runoff prediction in large-
sample hydrological dataset in China. Specifically, when using ERA5 precipitation data, the
median NSE of LSTM across 544 basins reaches 0.57 (NSE $\geq$ 0.55 is considered the threshold
for good performance (Knoben et al., 2019; Newman et al., 2015)). When using CN05.1
precipitation data, the median NSE for LSTM in regional modeling and PUB reached an
impressive 0.95 and 0.93, respectively. This phenomenon may be due to the runoff data product
used in this study, which is also simulated using CN05.1 data (as described in Section 4.1). The
use of CN05.1 precipitation data enables other subsequent models to demonstrate excellent
prediction performance, a point that will not be revisited in the following sections.
It is worth noting that under different precipitation data, when the PUB of 544 basins was
conducted using five-fold cross-validation, the prediction skills were comparable to those of
regional modeling. This indicates that the LSTM model also possesses strong generalization ability
in China's basins, achieving more accurate hydrological predictions in ungauged basins by learning
from hydrological data of other basins. Surprisingly, under the ERA5 precipitation data conditions,
the PUB results for some basins outperformed the prediction performance of regional modeling.
In theory, regional modeling uses data from all basins for training and is generally regarded as
providing superior predictions, especially for individual basins. When a basin's own data is
included in the training set, the model is expected to perform better in predicting that basin's runoff.
However, as illustrated in the scatter plot at the bottom of Figure 9, there are instances where the



PUB performance exceeds that of regional modeling when different precipitation data are used.
Notably, under the ERA5 conditions, the number of such basins is higher. This phenomenon may
indicate that the ERA5 precipitation data exhibits a certain degree of non-uniformity among
different basins (as shown in Figure 5), which affects the effectiveness of regional modeling. The
accuracy and spatial consistency of meteorological data jointly determine their impact on the
predictive performance of hydrological models. This not only influences the model's
representation of hydrological processes but may also significantly shape its generalization
performance. Therefore, we suggest that future related studies should comprehensively consider
the differences in training samples while also taking into account the accuracy of model input data.

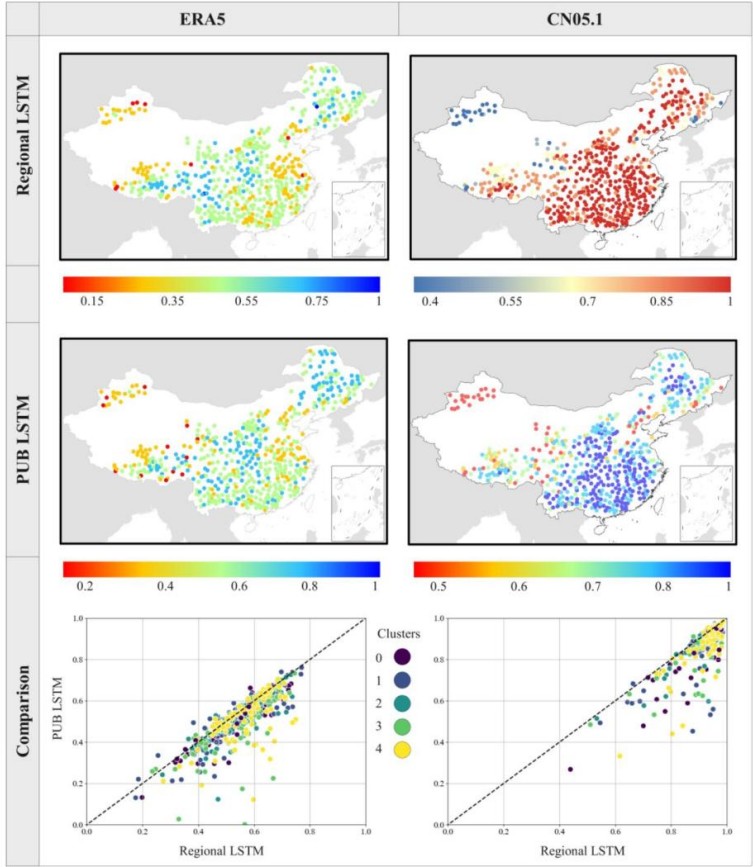




**Figure 9. LSTM models prediction performances using ERA5 and CN05.1 precipitation data.**

4.4 Performance and selection of hybrid models

Figure 10 shows the prediction performance of four alternative and differentiable hybrid hydrological models for regional modeling across 544 basins. When the model input is based on the CN05.1 dataset, the overall performance of the different hybrid models aligns closely with that of the pure LSTM model, and the spatial distribution characteristics of the NSE across the 544 basins are also highly similar. However, when the ERA5 precipitation data is used for regional modeling, the distribution of NSE varies among the basins. Although the median NSE of each hybrid model and the pure LSTM model across the 544 basins is generally consistent, coupling the process-based model (PBM) with LSTM results in a more stable prediction performance across different basins. This improvement may be attributed to the fact that the differentiable hybrid model follows to the law of water balance during the modeling process, avoiding the sacrifice of performance in more challenging basins in order to improve the overall performance across all basins. Therefore, the modeling strategy that couples PBM with deep learning, particularly the differentiable hybrid hydrological model, can not only maintain overall prediction accuracy but also enhance the spatial robustness of the model and balance prediction skills across different basins.



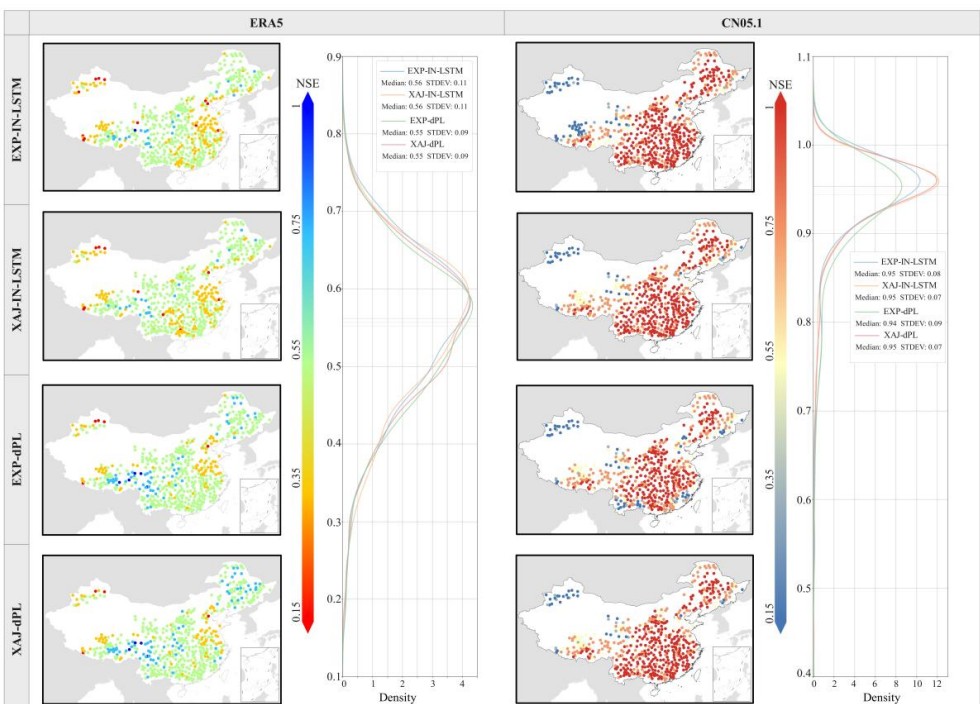

**Figure 10. Four hybrid models prediction performances using ERA5 and CN05.1 precipitation data.**

Further analysis of the specific performance of different basins under various hybrid models is presented in Figure 11 (a) and (b), which show the distribution of the optimal hybrid model for each basin based on two sets of precipitation data. In general, under the two precipitation datasets, CN05.1 and ERA5, the alternative hybrid modeling scheme—where the predicted values of the PBMs are used as an additional input for deep DL—exhibits superior performance across more basins than the differentiable hybrid modeling scheme, which involves parameterizing the process model and coupling it with DL. Although the optimal hybrid model for some basins remains consistent regardless of the precipitation data used as input, for most basins, changes in the input data lead to a shift in the most suitable hybrid model. This phenomenon indicates that the accuracy and reliability of meteorological data significantly influence the selection of hybrid models.





Figures 11 (c) and (d) show the distribution of the best hybrid models across various water
systems and climate regions. The results indicate that the Yellow River Basin and the Songliao
River Basin show a clear preference for alternative hybrid modeling methods. Notably, in the
Songliao River Basin (climate region 3), the EXP-INLSTM model outperforms others in a greater
number of basins. This may be attributed to the fact that the EXP-HYDRO model includes rain-
snow partitioning and snowmelt modules specifically designed to address snow storage, allowing
for a more accurate representation of the influence of snow on runoff. Additionally, in climate
region 6 (the high-altitude region of the Qinghai-Tibet Plateau), the EXP-dPL model demonstrates
strong suitability when using CN05.1 data. This area is influenced by the combined effects of
perennial snow and frozen ground, while the high-latitude climate region 3 is primarily impacted
by seasonal snow. This distinction indicates that although the hybrid modeling scheme coupling
EXP and LSTM generally outperforms the combination of XAJ and LSTM in basins affected by
snow storage, further subdivision reveals that different hybrid methods involving EXP and LSTM
may be better suited for basins with varying snow characteristics
Moreover, the optimal hybrid model for most basins is influenced by the source of
meteorological data. In humid regions, such as the middle and lower reaches of the Yangtze River
and the Southeastern Rivers, the hybrid models combining XAJ and LSTM generally outperforms
the hybrid models of EXP and LSTM across different precipitation datasets. Conversely, in the
Haihe River, the EXP and LSTM hybrid model consistently represents the best choice for most
basins. In other basins with varying climatic characteristics, the hybrid models of EXP and LSTM
demonstrate better predictive performance under ERA5 data, whereas the XAJ and LSTM hybrid
model performs better under CN05.1 data. This further confirms the impact of the accuracy and
reliability of meteorological data on the selection of hybrid hydrological models.








**Figure 11. Comparison of hybrid models performances across basins using ERA5 and**
**CN05.1 precipitation data.**

To evaluate the generalization ability of different hybrid hydrological models, the same PUB

scheme  for pure LSTM described in Section 4.3 was adopted to conduct 5-fold cross-validation

on 544 basins. The prediction performance of each hybrid model, using different precipitation data

as input, is shown in Figure 12. Overall, the prediction performance distribution of various hybrid

hydrological models across the 544 basins is relatively consistent under the same input data. With

ERA5 precipitation data, the hybrid model that performed best in the PUB is EXP-dPL, with a

median NSE of 0.55. In contrast, under CN05.1 precipitation data, XAJ-dPL demonstrates the best

generalization ability, achieving a median NSE of 0.95. It is worth noting that when different

meteorological data are used as input, the differentiable hybrid model exhibits better predictive

performance than the alternative hybrid model in the PUB. This indicates that the differentiable

modeling scheme can enhance the adaptability and generalization performance of the hybrid model

in ungauged basins.

The scatter plots in Figure 12 show the comparison of NSE values between regional modeling

and PUB for each basin. Under CN05.1 precipitation data, when XAJ-dPL was tested through 5-

564          fold cross-validation, the prediction performance of certain basins exceeded that of regional

modeling significantly. This phenomenon indicates that not all basins require their own historical

data for training when making hydrological predictions. In some cases, modeling based on the

hydrological information from other basins in the region can achieve better prediction results. This

further verifies the potential of hybrid modeling strategies, particularly differentiable hybrid

modeling methods, in enhancing model extrapolation capabilities and generalization performance.





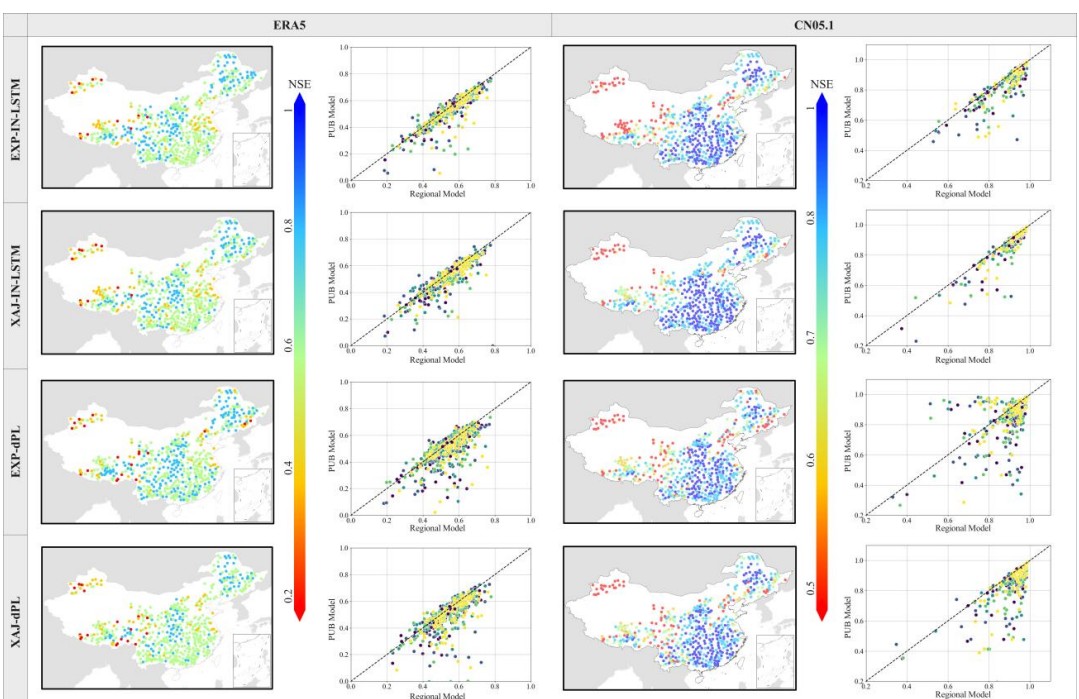

**Figure 12. Comparison of PUB performances of four hybrid models.**

4.5 Evaluation of water budget closure

To verify the physical consistency of the prediction results from different models, the water

budget closure and water imbalance ratio for 544 basins were calculated during the test period.

This study used a adapted long-term water budget for each basin (Tan et al., 2022; Wang et al.,

2014) to evaluate the basin water balance closure:

$$|P - ET - Q| = \varepsilon \qquad (7)$$

Where P is precipitation from ERA5 or CN05.1 dataset, ET is evapotranspiration (observations

provided by ERA5 or output of differentiable hybrid models), Q is runoff (observations or

predictions from different models), and ε is the water budget imbalance. The smaller the value of

ε, the better the water budget balance of the basin.

Figure 13 shows the difference of the annual water budget closure (ε) and the water imbalance

ratio (ε/P) for 544 basins among different models. The detailed data sources for runoff and



evapotranspiration associated with the various indicators in the figure are provided in Table S6 of
Supplementary Materials. The results show that when the simulated runoff from different hybrid
models replaces the observed values, the basin's water balance closure does not change
significantly. However, when the evapotranspiration data output by EXP-dPL replaces the original
ET data, the overall water imbalance across the 544 basins is significantly reduced. This
phenomenon indicates that, under the framework of EXP-dPL, the model's runoff output can not
only maintain high prediction accuracy but also conform better to the water budget constraint,
reflecting good physical consistency. It should be noted that the dataset used in this study has not
been strictly calibrated against actual observation sites, which may be one of the reasons why the
ET data output by XAJ-dPL caused a deterioration in water balance closure in some basins.
Nevertheless, the above results can still prove to a certain extent that the differentiable hybrid
modeling method can not only provide high-precision runoff predictions, but also output
intermediate hydrological variables that align with hydrological mechanisms. This ensures better
compliance with physical constraints while maintaining prediction performance. Therefore, the
differentiable hybrid modeling scheme represents a modeling approach that balances both physical
consistency and prediction skills.

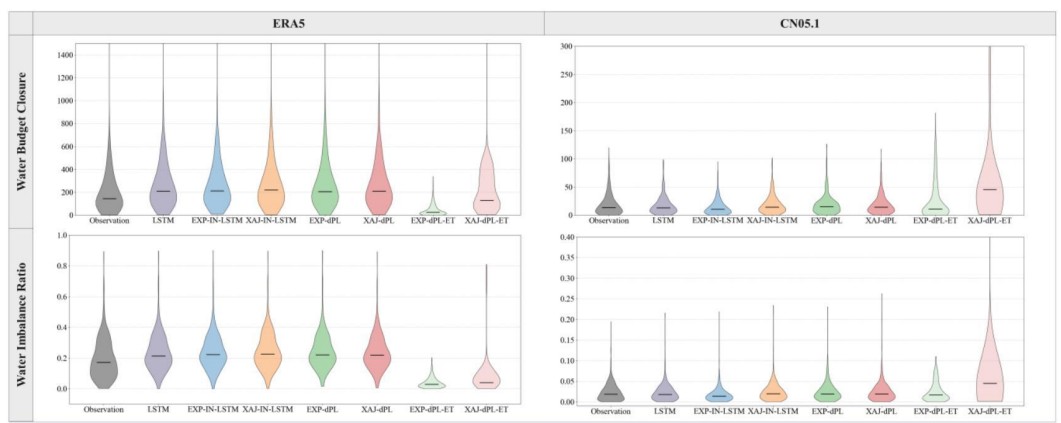




**Figure 13. Distribution of annual (a) water budget closure (ε) and (b) water imbalance ratio (ε/P) for all 544 basins during the test period.**

## 5. Conclusion

This study systematically evaluated the hydrological modeling of a large-sample hydrological dataset for 544 basins in China, analyzed the applicability of multiple hydrological models in complex hydrological environments, and provided a valuable basis for the selection and optimization of future hydrological models. Through a comparison of different hydrological modeling methods, the study reached the following three main conclusions:

(1) The accuracy and reliability of meteorological data have a key impact on the predictive performance of hydrological models. For most basins in China, changes in the source of meteorological data may lead to differences in the performance of similar models. Accurate precipitation data can help models better simulate the runoff generation process in the basin, thereby more effectively capturing the hydrological characteristics of the basin. During the selection and application of models, the quality and adaptability of meteorological data are crucial factors that determine the model's predictive capability.

(2) The hybrid modeling method shows strong predictive performance and generalization capabilities. In the prediction of ungauged basins, whether using ERA5 or CN05.1 precipitation data, the generalization ability of the hybrid modeling method surpasses that of the pure LSTM model. This indicates that the hybrid model is better equipped to adapt to the basins with different hydrological characteristics, providing more stable prediction results. This highlights the advantages of hybrid modeling methods in complex hydrological environments and their robust adaptability across various regions and climatic conditions.

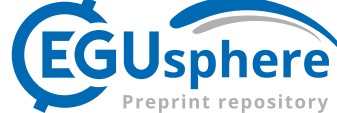

(3) Compared with alternative hybrid modeling schemes, the differentiable hybrid modeling

scheme achieves a deeper understanding of hydrological processes. Due to the seamless

coupling of process-based models, the differentiable hybrid hydrological model can

output unobserved intermediate hydrological variables. At the same time, the runoff

prediction results of this hybrid modeling scheme are more consistent with the water

budget balance at the basin scale, providing more comprehensive and accurate support

for hydrological prediction.

Although this study provides a relatively systematic evaluation of hydrological modeling in
China's basins, the meteorological and runoff data used have not been corrected, which may limit
the applicability of the dataset. Future studies should consider using observational data for
correction to enhance the dataset's reliability and further improve the accuracy of model
predictions. While this study primarily focuses on basins in China, it is hoped that the methods
and datasets presented in this research can serve as a reference for other regions lacking
observational data and promote similar hydrological modeling efforts worldwide.

**Data and Code Availability**

The dataset for 544 China basins in this study can be downloaded from
https://github.com/Yq-H47/Catchment-Attributes-and-Meteorology-dataset-for-China-544-
basins. And the models are available at https://github.com/Yq-H47.

**Declaration of competing interest**

No potential conflict of interest was reported by the authors.

**Acknowledgment**

This work was supported by the National Natural Science Foundation of China (grant
numbers 42371425, 42325107, 42330108) .



**CRediT authorship contribution statement**

**YH:** Conceptualization, Data curation, Formal analysis, Funding acquisition, Investigation, Methodology, Software, Visualization, Writing – original draft. **HL:** Conceptualization, Data curation, Formal analysis, Investigation, Methodology, Software, Validation, Writing – original draft. **CZ:** Conceptualization, Funding acquisition, Investigation, Methodology, Resources, Supervision, Writing – review & editing. **DS:** Funding acquisition, Resources, Supervision, Writing – review & editing. **BX:** Conceptualization, Investigation, Methodology, Resources, Supervision, Writing – review & editing. **MC:** Conceptualization, Funding acquisition, Investigation, Methodology, Resources, Supervision, Writing – review & editing. **WC:** Data curation, Visualization, Writing – original draft. **RL:** Writing – original draft, Writing – review & editing.

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
