# Peer review of "Exploring Diverse Modeling Schemes for Runoff Prediction: An Application to 544 Basins in China"

_EGUsphere, 2025_

## Author Comment (AC1)

**Author comments on CC1- egusphere-2025-1161**

***Comment 1:** Is there any reference or rationale for the determination of the watershed boundaries?*

**Response:**

Thank you for your comments. The determination of watershed boundaries is very important. Our previous manuscript did not provide a detailed description of our method for determining watershed boundaries. In fact, when we established the dataset, we used the watershed boundaries provided by the HydroSHEDS dataset. We then screened the watersheds based on their area and deleted the watersheds with an area of less than one thousand square kilometers. We have added a description of the method for determining watershed vector boundaries in the data section. The modified details are as follows:

"....

To delineate watershed boundaries consistently across China, we utilized the HydroSHEDS dataset (Hydrological data and maps based on SHuttle Elevation Derivatives at multiple Scales; Lehner et al., 2008), which provides high-resolution hydrologic information derived from SRTM elevation data. Based on the D8 flow direction scheme, the outlet of each basin was determined and used to extract the upstream contributing area. To ensure data consistency and model applicability, we performed area-based filtering and excluded small basins with an area less than 1,000 km². This threshold was chosen to reduce the influence of spatial resolution mismatch and potential errors in meteorological data aggregation. As a result, a total of 544 basins were retained, representing a wide range of hydrological and climatic conditions across China. The final basin boundaries are shown in Figure 1(a), and outlet locations are provided in Supplementary Figure S1.

...."

***Comment 2:** The abbreviations of the models in the article are very confusing. Please explain them uniformly in the appropriate place.*

**Response:**

Thank you for your reminder. Our study did use 7 different models. To make the article easier for readers to understand, we have declared an abbreviation for each model after it is mentioned for the

first time. At the same time, we have summarized the detailed information of all the models used in this study into a table (as shown in Table 3). This table will be added to the new manuscript. The specific table content is as follows:

"….

**Table 3** The details for all the models.

| Categories | Model | Model inputs | Training set | Testing set |
|---|---|---|---|---|
| process-based model | EXP-HYDRO | meteorological forcings | | |
| | Xin'anjiang | meteorological forcings | | |
| Deep learning model | LSTM | meteorological forcings, static basin attributes | | |
| alternative hybrid model | EXP-IN-LSTM | EXP-HYDRO predicted runoff, meteorological forcings, static basin attributes | October 1, 1975, to September 30, 1995 | October 1, 1995, to September 30, 2015 |
| | XAJ-IN-LSTM | Xin'anjiang predicted runoff, meteorological forcings, static basin attributes | | |
| differentiable hybrid model | EXP-dPL | meteorological forcings, static basin attributes | | |
| | XAJ-dPL | meteorological forcings, static basin attributes | | |

…."

*Comment 3: Line 284 describes the PUB test method. Why are the remaining 9 clusters used for training?*

**Response:**

Thank you for your careful review. We used 5-fold cross validation to test the generalization ability of the model when extrapolating ungauged basins. Specifically, we first randomly divided 544 basins into five clusters. Four of the five clusters were used as training sets, and the model was trained during the training period of these four clusters. Then the prediction performance of the model was tested using the remaining test period data. The purpose of this is to ensure that the model

has neither been exposed to any data from the test basin during training nor learned any data from the test basin during the test period. Ensure that the test of the model's generalization ability spans both space and time. There is a problem with our statement here, and we will describe the pub test scheme in more detail in the new manuscript. The modified details are as follows:

"….

The validation process is as follows: the model is trained using the training period data from the basins in four of the clusters, and its performance is validated on the test period data from the remaining a cluster.

…."

***Comment 4:*** *The authors claim that the differentiable mixed hydrological model can output unobserved intermediate hydrological variables, but there is no data to support this.*

**Response:**

Thank you very much for your valuable comment. One of the potential advantages of the differentiable hybrid hydrological model is that it can output relatively reliable intermediate hydrological variables under the premise of runoff as the target variable. In our original manuscript, we hoped to illustrate that the model can output reliable intermediate hydrological variables by calculating the closure of the water budget. Specifically, when the evapotranspiration output by the differentiable hybrid hydrological model is used to measure the water budget, the number of basins that achieve closed balance increases significantly. Therefore, we believe that the model can output relatively reliable intermediate hydrological variables on the basis of having ideal prediction performance, and its output evapotranspiration can also better conform to physical consistency, so the model can output relatively reliable intermediate hydrological variables.

However, as you mentioned, the reliability of the hydrological variables output by the model can be most directly illustrated by directly comparing the intermediate hydrological variables output by the differentiable hybrid model with the observed values. Therefore, we add a comparative analysis of the hydrological variables output by the model and the corresponding observed values (ERA5-Land). The specific results and analysis will be added to the new manuscript, as follows:

"….

4.6 Validation of Intermediate Hydrological Variables

To further assess the physical consistency and interpretability of the differentiable hybrid hydrological model, we evaluated its ability to reproduce intermediate hydrological variables that were not used as training targets. Specifically, we focused on three key variables: evapotranspiration, soil water, and snowpack. These variables were output by the differentiable hybrid model during the testing period (1995–2015), and compared with corresponding reanalysis estimates from the ERA5-Land dataset. Figure x illustrates the time series of the spatial average (across 544 basins) for the three variables. All time series were normalized prior to plotting to facilitate shape comparison. As shown, the outputs from the hybrid model (blue lines) closely follow the seasonal and interannual variations of the ERA5-Land estimates (orange dashed lines) across all three variables. This suggests that the model not only provides accurate runoff predictions, but also captures physically plausible hydrological states and fluxes. To quantify this consistency at the basin scale, the Pearson correlation coefficient between the output time series and ERA5-Land reference time series was calculated for each basin and each variable. The median correlation across all basins reached 0.71 for evapotranspiration, 0.64 for soil water, and 0.52 for snowpack.

These results demonstrate the hybrid model's ability to internally simulate key components of the hydrological cycle, even though these variables were never directly used in model training or supervision. The snowpack correlation, although slightly lower, can be attributed to the fact that some basins are located in warm regions with negligible snow accumulation. Moreover, uncertainties in the reanalysis-based "reference" values may also affect the observed correlations. Overall, these findings confirm that the differentiable hybrid modeling framework can generate physically meaningful intermediate states, further supporting its potential for interpretable hydrological modeling and process-informed regional analysis.

[Figure]

Figure x. Comparison between the outputs of the differentiable hybrid model and ERA5-Land reanalysis data for three intermediate hydrological variables (evapotranspiration, soil water, and snowpack) during the testing period (1995–2015). All values are normalized, and each curve represents the spatial average across 544 basins.

…."

***Comment 5:*** *What does the spatial distribution map in Figure 12 mean? A detailed explanation*

*should be given in the image caption.*

**Response:**

Thank you for your careful review and helpful suggestion. We agree that the caption of Figure 12 lacked sufficient detail to help readers interpret the spatial distribution maps. In the revised manuscript, we have clarified that the maps on the left side of each subpanel display the spatial distribution of NSE values under PUB test for each hybrid model, while the scatter plots on the right compare PUB and regional performances at the basin level. The colors of the dots represent the basin clusters. We have now updated the figure caption as follows:

"….

Figure 12. Comparison of PUB performances of four hybrid models (EXP-IN-LSTM, XAJ-IN-LSTM, EXP-dPL, and XAJ-dPL) under two meteorological datasets (ERA5-Land and CN05.1). For each model and forcing case, the left-side map shows the spatial distribution of the NSE values for each basin under PUB test, with warmer colors indicating lower NSE performance and cooler colors indicating higher performance. The right-side scatter plots compare PUB model performance (y-axis) to regional model performance (x-axis) for each basin. Each point represents a basin, and point colors correspond to different clusters. The dashed 1:1 line serves as a reference for comparing PUB and regional performances.

…."

*Comment 6: Why do you use the runoff predictions for water balance assessment? What is the purpose of calculating the water imbalance ratio? Please add an explanation.*

**Response:**

Thank you for your insightful reminder. In Section 4.5, we used the runoff predictions and evapotranspiration output by the differentiable hybrid model to calculate the water imbalance ratio. Because for basins with different dry and wet conditions, the imbalance degree of water conservation under the same error term is not exactly the same. For example, because the predicted runoff is 10 mm higher than the difference between precipitation and evapotranspiration, the reliability of the prediction results for basins with daily average precipitation of 50 mm and 1 mm is obviously far apart. Therefore, we chose to use the ratio of the error term to precipitation to measure the physical consistency of the runoff prediction results used in different basins and the

reliability of the model output evapotranspiration. The resulting enhanced water balance closure ability shows the potential of differentiable hybrid models in improving the understanding of regional water resources availability. We have followed your suggestion and added relevant explanations. The added explanations are as follows:

"....

The water balance closure is assessed based on the outputs of the differentiable hybrid models, which are able to output predicted runoff and evapotranspiration (ET). To evaluate the physical consistency of these predictions, the sum of model-predicted streamflow and ET is compared against total precipitation during the testing period. To ensure comparability across catchments with varying climatic conditions, the water imbalance ratio ($\varepsilon/P$) is adopted. This ratio is defined as the absolute difference between precipitation and the sum of streamflow and ET, normalized by total precipitation. This ratio reflects the extent to which model outputs satisfy the fundamental water balance constraint. Using this ratio enables a consistent and interpretable measure of physical plausibility across diverse hydrological regimes. A lower imbalance ratio indicates better water budget closure, suggesting that the model not only produces accurate runoff predictions but also maintains hydrological consistency in terms of mass conservation. This evaluation provides insights into the model's ability to simulate key hydrological variables in a physically meaningful way.
...."

**Comment 7:** *There is an error in the labeling of Figure 11. Two sub-figures (b) appear. Please modify them.*

**Response:**

Thank you for your valuable correction. We have corrected the subfigure labels to avoid duplication. The Sankey diagram previously labeled as Figure 11(b) has now been relabeled as Figure 11(c), and the subsequent subplots have been updated accordingly. The modified Figure 11 and its caption are as follows:

"....

[Figure]

**Figure 11. Comparison of hybrid model performances across basins using ERA5-Land and CN05.1 precipitation data**. (a) Spatial distribution of the best-performing hybrid models under ERA5-Land forcing. (b) Spatial distribution under CN05.1 forcing. (c) Sankey diagram showing the consistency of best-performing models across datasets. (d) Number of best-performing basins for each model in the 9 major river basins. (e) Number of best-performing basins in the 7 climate sub- regions.

…."

We would like to thank the editors and reviewers once again for their valuable suggestions on our manuscript. We have incorporated these suggestions into the revised manuscript. Looking forward to hearing from you.

Chunxiao Zhang

Corresponding author

E-mail address: zcx@cugb.edu.cn

References:

Gray, L. C., Zhao, L., & Stillwell, A. S., 2023. Impacts of climate change on global total and urban runoff. Journal of Hydrology, 620, 129352. https://doi.org/10.1016/j.jhydrol.2023.129352

Jiang, S., Zheng, Y., Solomatine, D., 2020. Improving AI System Awareness of Geoscience Knowledge: Symbiotic Integration of Physical Approaches and Deep Learning. Geophysical Research Letters 47, e2020GL088229. https://doi.org/10.1029/2020GL088229

Koya, S. R. , & Roy, T. . (2024). Temporal fusion transformers for streamflow prediction: value of combining attention with recurrence. Journal of Hydrology, 637.https://doi.org/10.1016/j.jhydrol.2024.131301

Kratzert, F., Klotz, D., Herrnegger, M., Sampson, A.K., Hochreiter, S., Nearing, G.S., 2019. Toward Improved Predictions in Ungauged Basins: Exploiting the Power of Machine Learning. Water Resources Research 55, 11344–11354. https://doi.org/10.1029/2019WR026065

Nai, C., Liu, X., Tang, Q., Liu, L., Sun, S., & Gaffney, P. P. J. (2024). A novel strategy for automatic selection of cross-basin data to improve local machine learning-based runoff models. Water Resources Research, 60, e2023WR035051. https://doi.org/10.1029/2023WR035051

Nearing, G., Cohen, D., Dube, V., Gauch, M., Gilon, O., Harrigan, S., Hassidim, A., Klotz, D., Kratzert, F., Metzger, A., Nevo, S., Pappenberger, F., Prudhomme, C., Shalev, G., Shenzis, S., Tekalign, T. Y., & Weitzner, D., 2024. Global prediction of extreme floods in ungauged watersheds. Nature, 627(8004), 559-563. https://doi.org/10.1038/s41586-024-07145-1

Zhan, Y. , Guo, Z. , Yan, B. , Chen, K. , Chang, Z. , & Babovic, V., 2024. Physics-informed identification of pdes with lasso regression, examples of groundwater-related equations. Journal of Hydrology, 638. https://doi.org/10.1016/j.jhydrol.2024.131504.

Lehner, B., Verdin, K., Jarvis, A., 2008. New Global Hydrography Derived From Spaceborne Elevation Data. Eos Trans. Am. Geophys. Union 89, 93–94. https://doi.org/10.1029/2008EO100001.

Tan, Xuejin, Liu, B., Tan, Xuezhi, Chen, X., 2022. Long-Term Water Imbalances of Watersheds Resulting From Biases in Hydroclimatic Data Sets for Water Budget Analyses. Water Resour. Res. 58, e2021WR031209. https://doi.org/10.1029/2021WR031209.

Wang, S., Huang, J., Li, J., Rivera, A., McKenney, D.W., Sheffield, J., 2014. Assessment of water

budget for sixteen large drainage basins in Canada. J. Hydrol. 512, 1–15. https://doi.org/10.1016/j.jhydrol.2014.02.058.

Luo Y, Ma N, Zhang Y. Divergent vegetation greening's direct impacts on land-atmosphere water and carbon exchanges in the northeastern Tibetan Plateau. Global and Planetary Change. 2025;251: 104825. https://10.1016/j.gloplacha.2025.104825.

Liu H, Wang J. Analyzing the impact of different forms of precipitation (rain and snow) on drought. Journal of Water and Climate Change. 2025;16(3):1119-1135. https://10.2166/wcc.2025.691.

Niu, Z., Liu, Y., Feng, L. et al. Spatio-temporal distribution characteristics of different types of compound extreme climate events in the Yangtze River Basin. Nat Hazards. 2025;121, 9261–9280. https://doi.org/10.1007/s11069-025-07165-8.

---

## Author Comment (AC2)

**Author comments on CC2- egusphere-2025-1161**

**Comment 1:** *Why did the author only choose two sets of precipitation data, or did the temperature data also come from two sets of data products? A more detailed description of the data source is needed.*

**Response:**

Thank you for your careful review and comments. In our study, we selected two sets of precipitation data and corresponding temperature data, mainly based on the following considerations:

1. Data accuracy and reliability: The two datasets we selected have been widely used in previous studies (Xie et al., 2025; Yu et al., 2025; Ma et al., 2024). Therefore, we believe that these data can provide a solid foundation for our research and provide effective reference for future research.

2. Purpose of comparative analysis: Using more data products for model comparison can indeed improve the credibility of research results. However, due to the length of the article, we selected two data sets in this study. In the future, we will refer to more data sets for more in-depth research.

3. Availability and coverage: We also considered the spatial resolution and temporal coverage of the data when selecting data. At the same time, in order to ensure the fairness of the comparison of different data, we need to consider the consistency of the start and end time of the selected data sets. After comprehensive consideration, we finally selected the ERA5-Land and CN05.1 data sets and ensured that their training time and test time were exactly the same.

We have further described the sources of the data in detail in the manuscript so that readers can understand our data selection process more clearly. The specific contents after modification are as follows:

"…

The meteorological data used in this study are sourced from the ERA5-Land and CN05.1 (Gao et al., 2013) datasets. Both datasets provide multiple meteorological variables, including daily precipitation and 2-meter air temperature, which are used consistently throughout this study. The full list of meteorological forcing elements and their corresponding units is shown in Table 1. These

two datasets were selected to extract basin-scale meteorological forcings based on the following considerations:

(1) ERA5-Land dataset: Although previous studies have noted that ERA5-Land data may exhibit certain deviations in East Asia, the dataset has several notable advantages. It offers a wide range of meteorological variables such as precipitation, temperature, radiation, humidity, and wind speed. Additionally, it spans a long historical period at a daily resolution, making it highly suitable for large-sample hydrological modeling across extended time scales.

(2) CN05.1 dataset: This gridded dataset is interpolated from over 2,400 national meteorological stations across China. In addition to precipitation, it provides high-resolution air temperature data, making it suitable for regional-scale climate analysis. Owing to its dense observational basis, CN05.1 is considered more accurate in reflecting local meteorological trends and spatial heterogeneity.

To ensure consistency, each model experiment used both precipitation and temperature from the same data source. ERA5-Land precipitation was used alongside ERA5-Land temperature, and similarly for CN05.1. These variables were aggregated to the catchment scale using area-weighted averaging. By incorporating both a global reanalysis product (ERA5-Land) and a regionally calibrated observational product (CN05.1), this study aims to evaluate the robustness of hydrological models under different meteorological forcing conditions, and to examine the sensitivity of model performance to data source selection. This dual-dataset strategy also provides useful insights for regions where observational data may be sparse or incomplete.
…"

*Comment 2: The description of the hybrid model structure in Section 3.4 is confusing. Please try to describe the operating logic of the two hybrid models separately.*

**Response:**

Thank you for your constructive feedback. We agree that the original description of the hybrid model structures in Section 3.4 could be clearer. In the revised manuscript, we have reorganized and rewritten this section to separately and more explicitly describe the operating logic of the alternative hybrid model and the differentiable hybrid model. The modified content is as follows:

"…

Both alternative hybrid modeling and differentiable hybrid modeling schemes are designed to combine the advantages of process-based models (PBMs) and deep learning (DL) models. The architectures of these two types of models are illustrated separately in Figure 4, and their working mechanisms are described below.

(1) Alternative hybrid modeling scheme

In this approach (Figure 4a), the PBMs (EXP or XAJ) are used first to simulate runoff based on meteorological inputs. The LSTM model then serves as a post-processing tool, taking both the PBM-simulated runoff and additional inputs—including the original meteorological forcing and static basin attributes as inputs. The LSTM is trained to learn and correct the discrepancies between the PBM outputs and the target runoff data. This method leverages the LSTM's ability to capture residual nonlinear relationships, thereby compensating for limitations of the PBMs in representing complex processes across large-sample and diverse basins.

(2) Differentiable hybrid modeling scheme

In this approach (Figure 4b), the discrete ordinary differential equations that define the hydrological processes in the PBMs are encoded directly into recurrent neural network (RNN) units, allowing the model to be fully differentiable. Static basin attributes are incorporated into the framework through a neural network-based parameterization scheme, which dynamically generates hydrological parameters. This allows the model to adapt the physical parameter values based on basin characteristics, overcoming the limitations of traditional PBMs that rely on fixed parameters. The model is trained end-to-end using backpropagation through time, enabling joint optimization of both hydrological states and parameters. In addition to discharge simulation, this framework also enables the output of intermediate hydrological variables, such as soil moisture and evapotranspiration, facilitating process-level interpretation and diagnostics.
…"

***Comment 3:*** *Line251: Which of the 6 categories the 15 attributes belong to needs additional explanation, or should be added to Table 2.*

**Response:**

Thank you for your suggestion. We neglected to explain the attribute categories of each watershed. Your suggestion is very valuable. It is more intuitive and clear to mark the attribute categories

directly in Table 2. We have revised Table 2 in the new manuscript. The specific content is as follows

"…

**Table 2** Static basin attributes data for 544 basins.

| Attribute | Categories | Description | Unit | Source |
|---|---|---|---|---|
| area | topography | Basin area | km2 | This study |
| srftopo | topography | Surface (rock + ice) elevation | m | Amante and Eakins (2009) |
| slope_avg | topography | Mean subgrid slope (inner slope) | m/m | Amante and Eakins (2009) |
| wcap | Soil | Maximum soil water capacity | $Kg/m^2$ | Hagemann and Stacke (2015) |
| wava | Soil | Plant available water | $Kg/m^2$ | Hagemann and Stacke (2015) |
| Fveg | vegetation | Fractional vegetation cover climatology relative to LSM | / | Hagemann (2002) |
| Lai | vegetation | Leaf area index | $m^2/m^2$ | Hagemann (2002) |
| p_mean | climate | Mean daily precipitation | m | This study |
| pet_mean | climate | Mean daily potential evapotranspiration | m | This study |
| aridity | climate | Ratio of Mean PET to Mean Precipitation | - | This study |
| frac_snow | climate | Fraction of precipitation falling on days with temp < 0 ∘C | - | This study |
| high_prec_freq | climate | Frequency of days with ≤ 5× mean daily precipitation | - | This study |
| high_prec_dur | climate | Average duration of high precipitation events | - | This study |
| low_prec_freq | climate | Frequency of dry days (< 1 mm/day) | - | This study |
| low_prec_dur | climate | Average duration of dry periods | - | This study |

…"

***Comment 4:*** *Line455: The author mentioned here the accuracy of climate characteristics and rainfall data. Among the 15 attributes, which meteorological data product is used to calculate "p_mean", "pet_mean", etc., or did the author use other methods?*

**Response:**

Thank you for your reminder. Regarding the watershed attribute data used in the study, we would like to make the following clarifications:

1. All watershed attribute data belonging to climate types are calculated from the meteorological time series of each watershed (1975.10.1-2015.9.30), including: "p_mean", "pet_mean", "aridity", " frac_snow", "high_prec_freq", "high_prec_dur", "low_prec_freq", "low_prec_dur". The calculation of these attributes is consistent with their descriptions.

2. There are actually two sets of climate-related watershed attributes, corresponding to the two datasets used in the study. When the data provided by ERA5-Land is used for model training

and testing, the watershed attributes are also calculated based on the ERA5-Land data; when the data provided by CN05.1 is used for model training and testing, the watershed attributes are also calculated based on the CN05.1 data (except "pet_mean" and "aridity").

3. Since the CN05.1 dataset does not provide potential evapotranspiration data, when the data provided by CN05.1 is used for model training and testing, "pet_mean" and "aridity" are calculated based on the ERA5-Land data.

To clarify this issue, we have added the following description in the 2.3 Static catchment attributes of the revised manuscript:

"…

In this study, catchment attributes related to climatic characteristics—specifically including p_mean, pet_mean, aridity, frac_snow, high_prec_freq, high_prec_dur, low_prec_freq, and low_prec_dur—were calculated from the corresponding meteorological forcing time series during the training and testing periods (1975/10/01–2015/09/30).To ensure consistency, we generated two sets of climate-related attributes, corresponding to the two meteorological datasets used in this study (ERA5-Land and CN05.1). When a model is trained and tested using ERA5-Land forcing, its associated catchment attributes are also calculated based on ERA5-Land data; similarly, when CN05.1 is used as the forcing, the climate-related attributes are calculated from the CN05.1 time series.

A specific exception is made for pet_mean and aridity, as the CN05.1 dataset does not directly provide potential evapotranspiration (PET). In these cases, PET-related attributes are consistently calculated based on the ERA5-Land dataset, even when CN05.1 is used for precipitation and temperature. This compromise ensures both the availability and consistency of these critical attributes while maintaining reasonable comparability between the two modeling configurations.

…"

***Comment 5:*** *The author uses the Budyko curve to examine the watershed water balance in Section 4.1, while in Section 4.5, the water budget closure method is employed. Why are different methods used to verify the watershed's water balance situation?*

**Response:**

Thank you for your careful review. As you said, we used different methods to measure the water

balance of the basin in Section 4.1 and Section 4.5. This is because we have different purposes in measuring the water balance at the basin scale in the two stages of the study, so we need to use appropriate methods. Specifically:

In Section 4.1, we further explored the differences and deviations between the two data sets after extracting the meteorological and runoff data of each basin in order to evaluate the number of basins that violate the water-heat balance when using different meteorological data. At this time, the meteorological and runoff time series used include the complete training period and test period (1975-2015).

The water balance evaluation work in Section 4.5 is mainly to verify the physical consistency of the prediction results of different models. Therefore, the runoff data used are the predicted values output by each model. At the same time, in order to avoid the unfairness of the absolute size of the water balance error terms of different dry and wet basins, we added the water imbalance ratio to measure the physical consistency of the prediction results of different models. At this time, the meteorological and runoff time series used only include the test period (1995-2015).

In order to better explain why we use different methods to measure water balance and avoid causing similar confusion to readers, we have added corresponding explanations in Sections 4.1 and 4.5. The modified details are as follows:

"…

4.1 Meteorological forcing assessment

…

The Budyko framework is employed   to assess the overall water balance behavior of each basin over the full period (1975–2015), based on precipitation, potential evapotranspiration, and runoff data. This analysis aims to evaluate the consistency and deviation patterns between meteorological forcing and runoff datasets under different data sources. Specifically, it provides a diagnostic tool for detecting basins with potentially unbalanced water budgets, which may indicate issues in either meteorological forcing or runoff simulations. Therefore, the Budyko curve here serves as a reference framework for identifying data-driven inconsistencies across basins and forcing products.

…

4.5 Evaluation of water budget closure

The water balance assessment in this section is focused on evaluating the physical consistency of

the model-predicted runoff during the testing period (1995–2015). The water budget closure analysis is used to compare precipitation, model-simulated runoff, and potential evapotranspiration for each model, aiming to quantify the degree of water balance closure in the model outputs. To account for differences in hydrological regimes, a water imbalance ratio ($\varepsilon/P$) is adopted as a metric to ensure comparability across basins. This approach provides insight into the process realism and hydrological plausibility of each model's predictions.

…"

**Comment 6:** *There are still many available high-quality meteorological data products. I can understand the author's decision to limit the scope of the article to control its length. However, this needs to be clarified in the conclusion section of the article.*

**Response:**

Thank you very much for your understanding and reminder. As you pointed out, there are many high-quality meteorological data products available. Our research selected two data products for relevant experiments and analysis after considering the time span and resolution. In the future, there will be more other high-quality meteorological data products. Therefore, our research focuses more on providing references by playing the role of cases. Your suggestion is very pertinent, and we have added an explanation of this point in the conclusion. The specific content is as follows:

"…

It is worth noting that although this study employs only two widely-used meteorological datasets, there exist many other high-quality meteorological products that could also support large-sample hydrology modeling. The choice to limit our analysis to these two products was based on considerations of data accessibility, spatial and temporal resolution, and the desire to maintain clarity and focus in model comparison. Future work may incorporate a broader set of meteorological forcings to further evaluate the robustness and generality of hydrological model performance across varying data sources. This study thus serves as an initial benchmark for such efforts in China, providing a reference framework for subsequent research.

…."

We would like to thank the editors and reviewers once again for their valuable suggestions on our

manuscript. We have incorporated these suggestions into the revised manuscript. Looking forward to hearing from you.

Chunxiao Zhang

Corresponding author

E-mail address: zcx@cugb.edu.cn

References:

Addor, N., Newman, A.J., Mizukami, N., Clark, M.P., 2017. The CAMELS data set: catchment attributes and meteorology for large-sample studies. Hydrol. Earth Syst. Sci. 21, 5293–5313. https://doi.org/10.5194/hess-21-5293-2017

Gray, L. C., Zhao, L., & Stillwell, A. S., 2023. Impacts of climate change on global total and urban runoff. Journal of Hydrology, 620, 129352. https://doi.org/10.1016/j.jhydrol.2023.129352

Jiang, S., Zheng, Y., Solomatine, D., 2020. Improving AI System Awareness of Geoscience Knowledge: Symbiotic Integration of Physical Approaches and Deep Learning. Geophysical Research Letters 47, e2020GL088229. https://doi.org/10.1029/2020GL088229

Kratzert, F., Klotz, D., Herrnegger, M., Sampson, A.K., Hochreiter, S., Nearing, G.S., 2019. Toward Improved Predictions in Ungauged Basins: Exploiting the Power of Machine Learning. Water Resources Research 55, 11344–11354. https://doi.org/10.1029/2019WR026065

Nearing, G., Cohen, D., Dube, V., Gauch, M., Gilon, O., Harrigan, S., Hassidim, A., Klotz, D., Kratzert, F., Metzger, A., Nevo, S., Pappenberger, F., Prudhomme, C., Shalev, G., Shenzis, S., Tekalign, T. Y., & Weitzner, D., 2024. Global prediction of extreme floods in ungauged watersheds. Nature, 627(8004), 559-563. https://doi.org/10.1038/s41586-024-07145-1

Xie Y, Kong D, Zhang Y, et al.Evaluation of the performance of multiple reanalysis forcing data in potential evapotranspiration estimation and its implication for actual evapotranspiration modeling.Journal of Hydrology[2025-07-07]. https://doi.org/10.1016/j.jhydrol.2025.133472.

Yu X, Zhang Q, Zeng X, et al.The distribution and driving climatic factors of agricultural drought in China: Past and future perspectives.Journal of Environmental Management, 2025, 377. https://doi.org/10.1016/j.jenvman.2025.124599.

Wang Z, Li M, Zhang X,et al.Prediction of long-term future runoff under multi-source data assessment in a typical basin of the Yangtze River.Journal of Hydrology: Regional Studies, 2024, 56(000). https://doi.org/10.1016/j.ejrh.2024.102053.

Ma X, Wang A. Evaluation and Uncertainty Analysis of the Land Surface Hydrology in LS3MIP Models Over China.Earth & Space Science, 2024, 11(7). https://doi.org/10.1029/2023EA003391.

Miao Y, Wang A. Evaluation of Routed-Runoff from Land Surface Models and Reanalyses Using Observed Streamflow in Chinese River Basins.Journal of Meteorological Research, 2020,

34(1):73-87. https://doi.org/10.1007/s13351-020-9120-z.

---

## Author Comment (AC3)

**Author comments on RC1- egusphere-2025-1161**

***Comment 1:*** *For assessing the performance of the different experiments, the authors compare the simulated streamflow to streamflow from VIC-CCN5.1... which is also simulated streamflow. This choice is justified, although I guess that VIC-CCN5.1 had to be evaluated against observed streamflow, so why not using it. To add to the confusion, several of the experiments come from models forced by CCN5.1. That induces a bias in the conclusions that can be drawn.*

**Response:**

Thank you for your careful review and comments. Your suggestion is very professional. For our research, using the observed runoff data of each basin as the target variable is the most rigorous. However, the reality is that we cannot obtain the daily runoff data of hundreds of basins in full. On the one hand, most of the runoff simulation studies in China are only conducted in specific areas, which is one of the reasons that prompted us to conduct large-sample hydrology studies in China. The standards for demarcating basin boundaries in these studies are not uniform, so even if we can obtain daily runoff data in full, we cannot guarantee that these data have good consistency. On the other hand, the data start and end times of the above studies are not the same, and even most of the time periods of the studies do not overlap, which is very unfavorable for the establishment of a large-sample basin data set. Therefore, when we first started data processing, we hoped to extract daily runoff data for hundreds of basins based on a relatively high-quality, relatively long time span daily runoff data set and a unified basin boundary demarcation standard.

Of course, we also fully agree with what you said that using runoff data products will lead to biased conclusions, so we originally planned to use the real hydrological station data of several basins to calibrate the runoff data products. However, the problem we face is that we only have real daily runoff data for 15 basins (the time span of each basin is about one year). And the above basins cannot cover different climate zones and water systems in space. If we only use these basins for data correction, it may lead to greater deviations. We think about it and make a cautious decision. Since it is impossible to fully guarantee that the data of each basin in the dataset is consistent with the actual observation under the existing objective conditions, we will try to simplify the data acquisition method on the basis of ensuring data consistency. This can ensure that the dataset has a

certain degree of availability and provide a reference for establishing a large-sample dataset in other areas that lack observation sites.

Regarding the issue of conclusion bias, we plan to add appropriate explanations in the Discussion section to ensure that readers can view our conclusions critically while understanding our goals. The relevant discussion added is as follows:

"…

A notable limitation of this study lies in the use of VIC-CN05.1 simulated streamflow as the reference for model evaluation. While this approach ensures nationwide spatial coverage and consistent hydrological boundaries across all 544 catchments, it may introduce systematic biases—particularly when evaluating models that are forced with the same meteorological dataset (CN05.1). The use of simulated rather than observed streamflow data could potentially favor certain models and compromise the neutrality of comparative performance assessments.

The primary rationale for adopting a runoff product instead of observational data stems from the limited accessibility and temporal inconsistency of observed streamflow records in China. Existing observed datasets often cover only specific regions, vary in spatial resolution and delineation standards, and exhibit non-overlapping time periods. These issues hinder the construction of a coherent, large-sample hydrological dataset with sufficient temporal depth and spatial uniformity. Given these constraints, the VIC-CN05.1 product was selected due to its relatively high simulation quality, long-term continuity, and compatibility with the CN05.1 precipitation product and unified basin boundaries.

Although this choice is methodologically justifiable, it is important to acknowledge the limitations it imposes on the interpretation of model performance. Comparative results may partially reflect consistency between inputs and evaluation targets rather than absolute predictive skill. Therefore, the findings should be interpreted with caution, particularly regarding the apparent superiority of models forced by CN05.1. Future work may consider integrating sparse but high-quality observed streamflow data for calibration or validation, thereby enhancing the robustness of the benchmark and supporting broader applicability in ungauged or data-scarce regions.

…"

***Comment 2 (About Abstract):*** *The abstract mentions the use of PBMs, but not what they are used for, neither what we can conclude about them. Line 36: conclusions about the two hybrid models are drawn, but those are not detailed before. L 40-42: This is not a concluding sentence for an abstract, this is the rationale of the study. Here we need you to give us the major guidance resulting from your work.*

**Response:**

Thank you for your reminder. Your suggestion on the abstract is very detailed. We have made the following changes to the abstract: added an explanation of the purpose of the process-based model; briefly introduced the operation of the two hybrid models (before summarizing the conclusion); added a concluding sentence at the end of the abstract, detailing the recommendations provided by this study. The content of the modified abstract is as follows:

"…

Hydrological modeling plays a key role in water resource management and flood forecasting. However, in China, with diverse geography and complex climate types, a systematic evaluation of different modeling schemes for large-sample hydrological datasets is still lacking. This study preliminarily constructs a dataset of catchment attributes and meteorology covering 544 basins in China, and systematically evaluates the applicability of two process-based models (PBMs: EXP-HYDRO model and Xin'an jiang model), long short-term memory (LSTM) models, and hybrid modeling methods. Among them, four hybrid models are developed: two process-based models are combined with the LSTM model using the alternative hybrid modeling scheme and the differentiable hybrid modeling scheme, respectively. The results demonstrate: (1) The accuracy of meteorological data critically impacts the prediction performance of hydrological models. High-quality precipitation data enables the model to better simulate the runoff generation process in the basin, thereby improving prediction accuracy. (2) The hybrid modeling method possesses regional modeling capabilities comparable to those of LSTM model. It also demonstrates strong generalization capabilities. In predicting ungauged basins, the hybrid model exhibits greater stability than the LSTM model. (3) Among the two hybrid modeling methods, the differentiable hybrid modeling scheme offers a deeper understanding and simulation of hydrological processes, along with the ability to output unobserved intermediate hydrological variables, compared to the alternative hybrid modeling schemes. Its prediction results are more consistent with the water

balance of the basin. The research results provide a systematic analysis for evaluating the applicability of different hydrological modeling methods in 544 basins in China, suggesting that high-quality meteorological data from consistent sources should be selected and considering the use of differentiable hybrid modeling schemes to better understand and simulate hydrological processes. This will help achieve higher prediction accuracy while ensuring the physical consistency of the prediction results.

…"

**Comment 3:** *Line 52: Why is the complexity of hydrological processes increasing? It seems to me that all this discussion is about natural processes, which do not complexify in time.*

**Response:**

Thank you for the valuable comment. We agree that natural hydrological processes themselves may not inherently become more complex over time. Our intention was to emphasize that, in recent years, the perception and modeling of hydrological processes have become increasingly complex, due to several factors:

1. Climate variability and change have led to more frequent extreme events (droughts, floods), making it harder to represent hydrological dynamics using traditional process-based models

2. Data limitations and heterogeneity, especially in ungauged basins or regions with sparse observations, increase the challenges in parameterizing and calibrating PBMs.

3. High expectations from stakeholders now require models to perform reliably under novel conditions or for diverse purposes, thus increasing the demand for more robust and generalizable modeling methods.

To avoid misunderstanding, we have revised the sentence as follows:

"…

However, with growing climate variability, and increasing demands on hydrological modeling, the perceived complexity and uncertainty in basin hydrological processes have increased, posing new challenges to the applicability of traditional process-based models (PBMs) in practice, especially under data-scarce and heterogeneous conditions.

…"

***Comment 4:*** *L 86: I do agree for physically-based models, but conceptual/empirical ones only need from 3 variables, namely precipitation, temperature and streamflow. This is not a substantial amount of high-quality data! For example, the EXP-HYDRO used by the authors exactly need these data, plus the day length, and the Xin'an jiang model only needs these data.*

**Response:**

Thank you for your insightful comment. We agree with your observation that conceptual and empirical hydrological models—such as EXP-HYDRO and Xin'an jiang—typically rely on a limited number of input variables (precipitation, temperature, runoff, and day length), and do not necessarily require a "substantial amount" of input data in terms of variable types.

Our intention, however, was to emphasize that even with a limited number of inputs, the reliability and performance of such models in practical applications often depend on the availability of continuous, high-quality observations, especially for streamflow data and meteorological drivers. Additionally, the calibration of model parameters can still involve subjectivity and nontrivial complexity, particularly when applied to basins with limited or noisy data.

To reflect this point more accurately and avoid confusion, we have revised the sentence as follows:

"…

Even when used to model hydrology for a single basin, such models often rely on the availability of continuous and reliable input data, and their parameterization can involve a degree of subjectivity and complexity, particularly in data-scarce or ungauged conditions.

…"

***Comment 5:*** *L 91: I do not agree, see previous comment*

**Response:**

Thank you for pointing this out. We understand your concern regarding the comparison between process-based models (PBMs) and data-driven models like LSTM.

We acknowledge that many conceptual models, such as EXP-HYDRO and Xin'an jiang, do not rely on highly detailed or fully physically-based parameters, and that their parameterization often combines empirical knowledge with simplified process representations.

Our intention was not to overstate the advantage of LSTM, but rather to highlight that deep learning models bypass the need for explicit physical parameterization, and instead rely on learning inputoutput relationships directly from data. However, we agree that LSTMs come with their own requirements, particularly the need for long, continuous, and high-quality historical datasets for effective training, which can also be a limitation in practice. To better reflect this balance, we have revised the sentence as:

"…

In contrast, deep learning models such as long short-term memory networks (LSTM) can learn the dynamic characteristics of basin hydrological processes from historical data and capture complex nonlinear relationships, without relying on explicit physical parameterizations. However, they typically require long-term, high-quality data for effective training, and their interpretability remains limited compared to PBMs.

…"

**Comment 6:** *L 168: From now onwards, I wonder if most elements should rather appear in the material and methods section of the manuscript*

**Response:**

Thank you for your constructive suggestion. We agree that much of the content in this paragraph—such as the dataset structure, variable types, model descriptions, and evaluation settings—would be more appropriately placed in the Data section. Our original intention was to briefly highlight the necessity and contribution of building a large-sample hydrological dataset covering diverse Chinese basins, as a key motivation for this study. However, we recognize that the inclusion of detailed technical information in the Introduction may affect the logical flow and clarity.

To address your comment, we have reorganized the manuscript structure:

The technical description of the dataset (number of variables, data sources, processing methods) has been moved to the Data section.

In the Introduction, we now briefly summarize the dataset's role and relevance in the study, without delving into implementation details.

**Comment 7:** *L 190: Why do accurate daily runoff observation data often need to be kept confidential?*

**Response:**

Thank you for your reminder. We acknowledge that the reasons for limited accessibility of daily

runoff observation data may not be immediately clear. In some countries, including China, hydrological data, particularly high-resolution runoff observations, are managed under strict institutional frameworks. Access to this data is often restricted for several reasons, including national regulations on water resource management, the perceived strategic importance of water data for flood control, water security, and infrastructure planning, as well as legacy data-sharing policies that allow agencies to maintain ownership and control over observation networks.

As a result, while some aggregated or monthly flow data may be available, high-quality daily discharge records are often difficult to obtain for research purposes or international sharing. Similar challenges have also been reported in other regions (e.g., South Asia, Africa and so on).

This situation underscores the importance of constructing a curated, consistent, and research-accessible dataset, as we have done in this study, to support comparative hydrological modeling and promote reproducibility.

We have clarified this point in the revised manuscript to provide better context. The specific contents are as follows:

"…

In China, obtaining datasets for large-scale hydrological studies is challenging for two main reasons. Firstly, access to accurate daily runoff observations is often restricted due to institutional regulations and data management policies.

…"

***Comment 8 (About Figure 1):***

*Figure 1:    Please make the different maps more uniform. Panel b uses a different color for foreign countries. In addition, please do not use the same color for China and seas (panel a). I also suggest removing the bottom right islands, as there as no basins there and they are originally not on the map. Imagine if French researchers put all French territories on all maps!!*

*Caption of Figure 1: In a I see the areas, in b the DEM, in c the catchments and in d the climates (only this ones correct). Please modify*

**Response:**

Thank you for the reminder. The use of different map backgrounds and inconsistent coloring may affect the visual coherence of the figure. In the revised manuscript, we have unified the background

color schemes across all sub-panels to improve clarity and comparability, and we have modified the colors of surrounding countries and seas to avoid potential confusion.

Regarding the inclusion of islands in the bottom-right corner of the map: we fully understand your concern. However, as this study is conducted using officially released national geographic data (from https://www.tianditu.gov.cn/), we are required to follow the standardized map representation guidelines mandated by relevant authorities. The inclusion of such elements is to comply with formal map-use conventions in Chian. We hope for your understanding in this matter.

In fact, several recent papers published in the journal focusing on China's hydrological research also used similar national base maps including the South China Sea region (as shown in Figs 1, 2, and 3).

[Figure]

**Fig 1. The Figure 2 from** *Assessing recovery time of ecosystems in China: insights into flash drought impacts on gross primary productivity* https://doi.org/10.5194/hess-29-613-2025

[Figure]

**Fig 2. The Figure 3 from** *The interprovincial green water flow in China and its teleconnected effects on the social economy* https://doi.org/10.5194/hess-29-67-2025

[Figure]

**Fig 3. The Figure 1 from** *Variation and attribution of probable maximum precipitation of China using a high-resolution dataset in a changing climate* https://doi.org/10.5194/hess-28-1873-2024

We have revised the figure caption accordingly to correctly describe the content of each sub-panel and avoid any previous confusion. The modified Figure 1 and its caption are as follows:

"….

[Figure]

**Figure 1. Spatial distribution of the 544 basins used in this study.** (a) Basin boundaries and areas. (b) Elevation distribution based on DEM data. (c) Divisions of China's nine major river systems and (d) seven climate regions (The map of China used in this study is from https://www.tianditu.gov.cn/.)

…"

***Comment 9:*** *L 234: I was completely lost here. There must be a nuance between the different terms (observation, runoff, runoff hydrograph), but I initially didn't get it. Only later on, while reading the results, I understood that the VIC-CN05.1 dataset is simulations from the VIC model forced by CN05.1. That was not clear at all.*

**Response:**

Thank you for your reminder. You are absolutely right to point out the importance of clearly distinguishing between observed and simulated runoff data. In this study, we used a national gridded runoff dataset—VIC-CN05.1, which was produced by driving the VIC (Version 4.2.d) hydrological model with the CN05.1 meteorological forcing. The CN05.1 dataset itself is based on interpolated

observations from more than 2400 stations across China, and the resulting VIC-CN05.1 runoff dataset provides 0.25° × 0.25° daily runoff estimates for the period 1961–2017.

We fully acknowledge that this runoff dataset is simulated, not directly observed. However, due to institutional constraints and the limited public availability of high-resolution daily runoff observations in China, VIC-CN05.1 has been widely adopted in large-scale hydrological studies as a proxy or substitute for runoff observations. If possible, in the future, we will definitely attempt to update the dataset with all runoff and meteorological data using observational data.

Recognizing the limitation of using simulated data, we conducted a comparison between the VIC-CN05.1 runoff series and actual observed streamflow records from 15 gauged basins with similar boundaries (see Supplementary Figure S2). The results show that, although not strictly calibrated, the simulated runoff series closely capture the overall temporal variation and seasonal trends of observed runoff, suggesting that the dataset is reasonably suitable for large-sample hydrological modeling.

We also acknowledge that constructing a fully observation-based hydrological dataset across hundreds of basins with consistent boundary delineation and time coverage is currently infeasible in China, due to limited public access to daily streamflow data. Thus, our study aims not to deliver a high-precision observational dataset, but rather to build a relatively comprehensive and internally consistent dataset that enables comparative and reproducible evaluation of hydrological models.

In light of your comment, we have revised the manuscript to avoid mislabeling simulated data as "observed." Instead, we now refer to the VIC-CN05.1 runoff data as a proxy for observations, and clearly acknowledge that it is model-simulated data used as an observation substitute due to the unavailability of direct measurements. We hope this clarification preserves the intent of our study while improving transparency and accuracy.

In the revised manuscript, we clarified the origin and nature of the VIC-CN05.1 dataset in the Data section and explicitly stated the purpose and findings of the comparison with observed data in Supplementary Figure S2. The specific contents are as follows:

"…

Due to the limited accessibility of daily observed runoff records in China, this study uses the VIC-CN05.1 dataset as a proxy runoff dataset. This dataset was generated using the VIC model driven by the CN05.1 meteorological forcing, and provides daily runoff estimates at 0.25° × 0.25°

resolution. Although this dataset is not based on direct streamflow observations, it has been widely adopted in previous studies and offers a physically consistent, nationwide runoff product. In this study, we treat it as a substitute for observed runoff in basins where actual measurements are unavailable. To validate its feasibility, runoff time series from 15 gauged basins were compared with actual streamflow records from hydrological stations (see Supplementary Figure S2). The results suggest that VIC-CN05.1 data can reasonably capture seasonal patterns and interannual variability, making it a suitable alternative for large-sample hydrological model evaluation in data-scarce regions.

…"

**Comment 10:** *L 284: Do you mean 4? There are 5 clusters*

**Response:**

Thank you for catching this typographical error. The manuscript should refer to four clusters, not nine. We have corrected the sentence to accurately describe the 5-fold cross-validation procedure. The specific contents are as follows:

"… the model is trained using the training period data from the basins in four of the clusters, and…"

**Comment 11:** *Figure 4: While a is understandable, I do not get b at all. What is FCNN? It is never defined in the text. Please improve or develop the caption.*

**Response:**

Thank you for pointing out that "FCNN" was not defined in the figure or the main text. In fact, FCNN stands for Fully Connected Neural Network, which serves as the parameterization channel: it takes static basin attributes (e.g., soil, terrain metrics) as inputs and maps them to the hydrological model parameters θ. We have now (1) added the definition of FCNN in the caption and main text, and (2) enriched the caption to make panels (a) and (b) fully self-contained. The specific content after the modification is as follows:

"…

[Figure]

**Figure 4. Structure of the hybrid hydrological models.** (a) In the conventional hybrid scheme, a process-based hydrological model (PBM) produces simulated runoff $Q_{sim}$, which—together with meteorological forcings (P: precipitation; T: temperature; Srad: solar radiation; VP: vapor pressure; $L_{nay}$: day length) and static catchment attributes—is fed into a data-driven network (LSTM + FCNN) to predict runoff Q. (b) In the differentiable hybrid scheme, the PBM's discrete equations are embedded into RNN units, while a Fully Connected Neural Network (FCNN) parameterization channel maps catchment attributes to the model parameters $\theta_p$. The entire architecture (parameters within both RNN and FCNN) is then optimized jointly via back-propagation, allowing hydrological parameters to vary adaptively across basins and climatic regimes.

…

In the differentiable hybrid modeling scheme (Figure 4b), standard recurrent neural network (RNN) units encode the discrete ordinary differential equations of the process-based hydrological model, ensuring mass balance and fundamental process representation. At the same time, we introduce a parameterization channel implemented as a Fully Connected Neural Network (FCNN), which ingests static basin attributes and produces the spatially varying parameter vector $\theta_p$. By jointly optimizing both the RNN weights and the FCNN parameters via back-propagation, the model can dynamically adjust its physical parameters conditioned on basin characteristics, overcoming the

fixed-parameter limitation of traditional PBMs and enabling cross-basin generalization.

…"

***Comment 12 (About Figure 5):***

*Figure 5: Please use the same range for the distribution of P values for the two products over the diverse basins. Also make sure to use the same categories, it seems that there are many more categories for CN05.1 than for ERA5. I guess this is basin-averaged P and T? Please specify.*

*Figure 5: The scale indicates a gradual color scale for P and T, but the maps only display categorical values, with only 5 colors. Please correct. What is the period? Is it the total period or the evaluation period (1995-2015)? These two comments are valid for most figures that follow*

**Response:**

Thank you for pointing out the potential confusion between the histogram scales and the map legend. To clarify our presentation—and without altering the original figure content—we have made the following changes:

1. Clarified the time period: In the revised caption we now explicitly state that all basin-averaged values are calculated over the full study period (from October 1, 1975, to September 30, 2015).

2. Explained the categorical coloring: Although the histograms use continuous bins to show the full distribution, the maps intentionally use five discrete categories to highlight broad hydro-climatic classes across China. These categories were chosen based on natural breaks in the combined ERA5-Land & CN05.1 distribution (<500 mm, 500–1000 mm, 1000–1500 mm, 1500–2000 mm, >2000 mm for precipitation), and similarly spaced for temperature. We have revised the figure legend and added a sentence to the caption to make this explicit.

3. Unified legend ranges: We confirmed that both the ERA5-Land and CN05.1 datasets share the same class boundaries: red consistently represents the highest precipitation category, while blue indicates the lowest. For temperature, green represents the lowest values, and red represents the highest. This alignment ensures direct comparability, even though the underlying datasets have slightly different numerical ranges.

We trust these changes resolve the inconsistencies and improve the clarity of our data presentation. At the same time, we have thoroughly checked all the figures in the manuscript and made

modifications to the legends and figure captions to ensure that readers can clearly understand the meaning of each figure. The modified Figure 5 and its caption are as follows:

"…

[Figure]

**Figure 5. Spatial distribution of five hydro-climatic categories of basin-averaged precipitation and temperature (1 Oct 1975–30 Sep 2015).** (a) Mean annual precipitation (mm): Top histograms display the continuous distribution of basin means. Maps use five discrete classes: <500, 500 to 1000, 1000 to 1500, 1500 to 2000, and >2000 mm, applied uniformly to both ERA5-Land (top row) and CN05.1 (bottom row). (b) Mean daily temperature (°C): Top histograms display the continuous distribution of basin means. Maps use five discrete classes: <5, 5 to 10, 10 to 15, 15 to 20, and >20 °C, applied identically to both datasets.

…"

***Comment 13:*** *L 407: How is the drought index calculated?*

**Response:**

Thank you for your reminder. In the revised manuscript, we have added a clear definition of the drought index and its calculation. The details are as follows:

"…

Figure 6 shows a scatter plot of the evaporation index (EI, the ratio of annual average evapotranspiration to annual average precipitation) and the drought index (Aridity, the ratio of annual average potential evaporation to annual average precipitation).

…"

***Comment 14:*** *L 415: That definitely induces a bias! It is easier to reproduce streamflow obtained from a model forced by a dataset, when you use the same dataset…*

**Response:**

Thank you for your reminder. You are absolutely right that using the same meteorological forcing (CN05.1) both to generate our "proxy" runoff via VIC and then to drive other models can introduce a positive bias in cross-model comparisons. Our primary objective, however, is not to report absolute predictive skill but to perform a relative evaluation of different modeling approaches under identical forcing conditions. By holding the input data constant, we ensure that differences in performance arise from model structure rather than from differences in meteorological inputs. We added relevant explanation in the Methodology to clarify this point:

"…

While the use of the same CN05.1 forcing to generate VIC-simulated runoff and to drive all subsequent models may inflate apparent performance, this design was chosen to isolate the effect of model formulation by eliminating variability in meteorological inputs.

…"

***Comment 15:*** *L 416: This is methods, not results*

**Response:**

Thank you for your correction. We agree that the description of the VIC-CN05.1 product belongs in

the Methods section rather than in Results. Accordingly, we have moved the sentence "…the runoff data product used in this study was simulated by the VIC model, which uses CN05.1 meteorological data. …" into the Methods section.

**Comment 16:** *L 420-425: This is discussions, not results*

**Response:**

Thank you for your correction. These statements should indeed be placed in the Discussion section rather than the Results section. In the revised manuscript, we have made several corresponding adjustments.

We have separated the Results and Discussion into distinct sections. The Results section now focuses solely on descriptive findings, such as the number of basins violating balance under each forcing. In contrast, interpretative content including the implications for water–energy closure and recommendations for future dataset construction has been relocated to the new Discussion section. Additionally, we have expanded the Discussion to provide deeper insights, addressing your earlier observation that our interpretation was too brief. Specifically, we now discuss the critical importance of ensuring mass–energy closure at the watershed scale across diverse climatic regimes. We also explore how calibration strategies, such as multi-objective optimization of flow and energy fluxes, can help reduce balance violations. Furthermore, we try to discuss the specific challenges and solutions for high-altitude, humid basins where snow processes and significant latent heat fluxes play a dominant role. The specific discussion content corresponding to this point is as follows: "...

The marked decrease in balance-violating basins under CN05.1 forcing highlights the critical role of accurate precipitation and energy inputs in large-sample hydrological datasets. Ensuring mass and energy closure is especially challenging in humid, high-altitude basins where snow accumulation, melt dynamics, and evapotranspiration interact strongly.

To mitigate balance errors, future dataset-building efforts could incorporate multi-objective calibration routines that jointly optimize streamflow, snowmelt timing, and energy fluxes. For high-altitude watersheds, integrating remote-sensing snow cover, station-based radiation corrections, and physically based snowpack models may further improve closure and data fidelity.

…"

***Comment 17:*** *Figure 6: what is the blue shaded area?*

**Response:**

Thank you for your reminder. The translucent blue band in both panels is intended to highlight the relatively "humid" basins. Specifically, basins where the aridity index (mean PET / mean P) is less than 1.5. In our original caption this feature was not described, which understandably caused confusion. The revised Figure 6 caption as follows:

"…

**Figure 6. Water balance for 544 basins, illustrated in a Budyko scheme for ERA5-Land (a) and CN05.1 (b).** Markers are coloured by the basin mean elevation. The translucent blue band marks the relatively humid regime (aridity index < 1.5).

…."

***Comment 18:*** *L 433: This is a somehow unfair comparison, as the reference data used to calculate NSE comes from VIC forced by CN05.1. Then, when you compare models forced by ERA5 to these data, you include the error coming from the PBM and the error coming from the input data set.*

**Response:**

Thank you for your timely correction. You are correct that, by using VIC-CN05.1 as the reference "observed" hydrograph, the NSE computed for models driven by ERA5-Land reflects both the structural error of each model and the mismatch between ERA5-Land and CN05.1 forcings.

To make this explicit, we have corrected this sentence in the revised manuscript and added relevant discussion. The specific content is as follows:

"…

Prediction performance (Nash–Sutcliffe efficiency) of both PBMs was higher when using CN05.1 precipitation than when using ERA5-Land precipitation; however, because CN05.1 was also used to generate the reference VIC-simulated runoff, this apparent improvement includes both model structural error and reduced forcing-mismatch error.

**Relevant discussion**

It should be noted that, by using VIC-CN05.1 simulations as the reference hydrograph, models

forced with ERA5-Land incur not only structural discrepancies relative to VIC, but also additional error arising from differences between ERA5-Land and CN05.1 inputs. In contrast, models driven by CN05.1 avoid the latter source of error. Consequently, the higher NSE observed under CN05.1 should be interpreted as the combined effect of more consistent meteorological inputs and model structural performance, rather than as a pure indicator of intrinsic model skill. Future work employing independent observed streamflow records will be required to disentangle these two components.

…."

**Comment 19 (About Figure 7):**

*Figure 7, left: what is this scale? It does not include regular intervals between values*

*Figure 7, caption: the authors state that the colormap include vales from 0 to 1. That would be great, to compare the four maps together. Unfortunately, the left maps do not use the same range as the right maps*

**Response:**

Thank you for your correction on the standardization of Figure 7. In the ERA5-Land-forced runs (left column), several basins yield negative NSE, whereas most CN05.1-forced runs (right column) have NSE ≥ 0. To preserve visibility of poor performance under ERA5-Land, the left maps are plotted over the range [–0.5, 1], while the right maps are restricted to [0, 1].

We have redrawn the legend and revised the caption to make this explicit. The specific contents are as follows:

"…

[Figure]

**Figure 7. Performance of process-based hydrological models during the testing period (1995.10.1–2015.9.30).** Spatial distributions of Nash–Sutcliffe efficiency (NSE) for (top row) EXP-HYDRO and (middle row) Xin'anjiang (XAJ) models under two precipitation forcings: ERA5-Land (left) and CN05.1 (right).

…."

*Comment 20: L 411 and following: The differences should be discussed in terms of what processes are important for these basins and what is the link with the processes present in the PBMs. We need interpretation!*

**Response:**

Thank you for this insightful suggestion.

It seems there might have been a slight misunderstanding regarding the line number. Based on the order in which you suggested, we guess that you meant line 441. We will take line 441 as an example for the following modification. If this is not correct, please feel free to let us know.

We have expanded the Discussion to link the observed model-selection patterns to key hydrological processes in each basin and to the mechanistic formulations within EXP-HYDRO (EXP) and Xin'an jiang (XAJ). Specifically, we added a new subsection titled "5.x Process-based Drivers of Model Preference" has been added, incorporating the rewritten content from lines 441 and following into this subsection, thereby integrating the new interpretation. The specific contents are as follows:

"…

5.x Process-based drivers of model preference

The differing strengths of EXP-HYDRO (EXP) and Xin'an jiang (XAJ) across China reflect their handling of soil moisture dynamics and runoff generation under distinct climate regimes. In the humid lowlands of the mid–lower Yangtze and southeastern rivers, deep soil moisture storage and vegetation-controlled evapotranspiration govern streamflow seasonality. XAJ's multi-layer soil moisture accounting and temperature-index evapotranspiration routine effectively reproduce the gradual release of baseflow and ET peaks, yielding consistently higher skill than EXP regardless of precipitation source. Conversely, in more arid northern basins such as the Haihe River system, rapid runoff via infiltration-excess and shallow subsurface flow is dominant. Here, EXP's explicit representation of infiltration, percolation, and evaporation processes aligns more closely with observed quick-flow responses, so EXP outperforms XAJ even when the forcing dataset changes.

Although EXP includes energy-balance routines for snowmelt, XAJ does not perform rain–snow separation but directly routes total precipitation into its soil and runoff modules. As a result, XAJ's robustness in snow-affected catchments stems from its simpler reliance on temperature-driven ET and multilayer storage, rather than on explicit snow physics. These patterns highlight that model preference depends not only on climatic setting—humid versus arid, lowland versus montane—but also on whether a model's structural emphasis (multilayer soil storage versus explicit infiltration-excess) matches the basin's dominant hydrological processes.

…."

***Comment 21:*** *L 469: The fact that the LSTM performs very well with CN05.1 comes from the fact that the authors do not try to reproduce observed streamflow but simulated streamflow. This means that LSTM does not excels in reproducing the processes leading to streamflow from meteorological input, but rather excels in mimicking the behavior of the VIC model. This is highly different and is caused by the experiment setup. In addition, this might indicate that the LSTM cannot cope with input errors*

**Response:**

Thank you for highlighting this important point. Your suggestion is correct and professional. Because the LSTM is trained to reproduce the VIC-CN05.1 simulated runoff, its high NSE under CN05.1 forcing largely reflects its ability to mimic VIC's behavior rather than to reconstruct the true physical processes leading to streamflow. This experimental setup therefore inflates apparent LSTM performance and masks its sensitivity to input errors.

We have revised this sentence and added a corresponding explanation to the Discussion section to clarify that the LSTM skill reported under CN05.1 is conditional on the simulated target and underscore the need for independent validation against real observations. The specific contents are as follows:

"…

When using CN05.1 precipitation data, the median NSE for LSTM in regional modeling and PUB reached 0.95 and 0.93, respectively. However, because the target hydrographs are themselves VIC-CN05.1 simulations, these high values primarily indicate LSTM's capacity to emulate the VIC model outputs, rather than its standalone process-learning skill.

…

…

Relevant discussion:

5.x Implications of learning from simulated targets

The exceptional NSE achieved by LSTM under CN05.1 forcing arises from training the network on VIC-simulated runoff. While this demonstrates the LSTM's flexibility in capturing the input–output mapping of a given process model, it does not necessarily imply proficiency in learning the underlying physics of runoff generation. Moreover, this setup can obscure the LSTM's

vulnerability to input biases. When driven by ERA5-Land, which differs more substantially from the VIC-CN05.1 climate statistics, the LSTM performance declines markedly, revealing its dependence on consistent forcing. To assess true hydrological generalization, future work should train and evaluate LSTM models against independent observed streamflow records and beyond the bounds of a single process model's behavior.

…."

**Comment 22:** *L 488-491: these are discussions, not results*

**Response:**

Thank you for noting that these statements are interpretative rather than strictly results. In the revised manuscript, we have taken your advice to clearly separate descriptive results from interpretative discussions by restructuring the sections. The "Results" and "Discussion" are now presented as two distinct sections. Additionally, all interpretative sentences, such as "This not only influences the … model input data," have been moved to the Discussion section.

In response to earlier feedback regarding the sparse discussion, we have enriched this part by elaborating on how the quality of input data and differences in samples affect both process representation and model transferability, as detailed in the new subsection 5.x:

"…

5.x Influence of Forcing Quality on Model Generalization

The accuracy and spatial consistency of meteorological forcing critically shape hydrological model performance and their ability to generalize. When inputs faithfully represent orographic precipitation patterns and energy fluxes—as in CN05.1—both process-based and data-driven models reproduce runoff dynamics more reliably. Conversely, mismatches or biases in precipitation phase, timing, or intensity introduce systematic errors that propagate through model components, degrading skill and transferability across basins. Future large-sample studies should therefore not only ensure balanced sampling of hydroclimatic regimes but also rigorously assess and, where possible, correct input data quality before model calibration and comparison.

…."

**Comment 23:** *Figure 9, 10: random scales prevent from comparing the different parts of the figure*

**Response:**

Thank you for noting that the use of different axis and colorbar limits can make cross-panel comparison difficult. In our original figures, each row's scale was chosen to best display the full spread of NSE values or density peaks for that particular model, but we recognize this hamper direct visual comparison across models and forcings. We have restructured the figure legends and made them clear in the captions so that readers can interpret each panel correctly. The revised Figures 9 and 10 and their captions are as follows:

"…

[Figure]

**Figure 9. Performance of LSTM models using ERA5 (left column) and CN05.1 (right**

**column) precipitation during regional modeling and PUB testing.** Top row (maps): Spatial

distribution of NSE during regional modeling. Middle row (maps): Spatial distribution of NSE

during PUB testing. Bottom row (scatter): Basin-by-basin comparison of PUB NSE (vertical axis)

vs. regional NSE (horizontal axis). Points are colored by clusters. The axes both span [0, 1].

[Figure]

**Figure 10. Four hybrid models prediction performances using ERA5-Land (left) and CN05.1**

**(right) precipitation data.** Spatial maps: Colorbars cover ranges of NSE values for different

models. Density plots: x-axes display the NSE values; y-axes are density.

…."

***Comment 24:*** *L 528-537: these are discussions, not results*

**Response:**

Thank you for noting that the explanations of EXP-HYDRO's snow handling and EXP-dPL's

suitability in the Qinghai–Tibet Plateau are interpretative rather than strictly "results." In the revised

manuscript, we have made adjustments. We have separated the Results and Discussion sections. All

descriptive performance metrics, including NSE values and model rankings, now reside in the

Results section. On the other hand, interpretative statements such as the reasons behind EXP-

HYDRO's improved rain–snow partitioning and its positive impact on snow-affected runoff, as well as the factors contributing to EXP-dPL's performance in high-altitude regions have been relocated to a new subsection in the Discussion. Furthermore, we have deepened the Discussion by expanding on the interpretation of snow processes. We explain how EXP-HYDRO's energy-balance snowmelt and partitioning routines effectively capture storage and melt dynamics. Additionally, we highlight how the improved accuracy of precipitation phase in CN05.1 enhances these effects. Our discussion also addresses how EXP-dPL's differentiable parameter channel is tailored to the unique energy and precipitation regimes of the Qinghai–Tibet Plateau, enabling it to outperform other models when driven by CN05.1 forcing. The specific content is as follows:

"…

5.x Snow-process representation and high-altitude performance

EXP-HYDRO's explicit rain–snow separation and energy-balance snowmelt modules store winter snowfall and release it based on temperature changes, yielding a more accurate runoff response in snow-dominated basins. CN05.1's station-interpolated precipitation better resolves snowfall events and snow–rain transitions than ERA5-Land, which explains EXP-HYDRO's improved NSE under CN05.1 forcing.

In the high-altitude Qinghai–Tibet Plateau (climate region 6), EXP-dPL further benefits from its differentiable parameterization channel: by learning spatially varying thermal degree-day factor and other relevant hydrological parameters directly from static attributes and CN05.1 inputs, it dynamically tailors its snow and runoff routines to local conditions. This flexibility leads to superior performance in this challenging environment compared to fixed-parameter PBMs.

…."

**Comment 25:** *Figure 12, 13: fonts are too small, we cannot read*
**Response:**

Thank you for your reminder. We have increased the font sizes in both Figures 12 and 13 (including axis labels, tick labels, legends, and colorbar annotations) to ensure their readability. All text elements now use at least an 8 pt font. At the same time, we carefully checked the font sizes of other figures in the manuscript and adjusted the fonts that were too small. We hope these adjustments can address your concern.

We would like to thank the editors and reviewers once again for their valuable suggestions on our manuscript. We have incorporated these suggestions into the revised manuscript. Looking forward to hearing from you.

Chunxiao Zhang

Corresponding author

E-mail address: zcx@cugb.edu.cn

References:

Addor, N., Newman, A.J., Mizukami, N., Clark, M.P., 2017. The CAMELS data set: catchment attributes and meteorology for large-sample studies. Hydrol. Earth Syst. Sci. 21, 5293–5313. https://doi.org/10.5194/hess-21-5293-2017

Kratzert, F., Klotz, D., Herrnegger, M., Sampson, A.K., Hochreiter, S., Nearing, G.S., 2019. Toward Improved Predictions in Ungauged Basins: Exploiting the Power of Machine Learning. Water Resources Research 55, 11344–11354. https://doi.org/10.1029/2019WR026065

Lu, M., Sun, H., Yang, Y., Xue, J., Ling, H., Zhang, H., and Zhang, W., 2025. Assessing recovery time of ecosystems in China: insights into flash drought impacts on gross primary productivity, Hydrol. Earth Syst. Sci., 29, 613–625, https://doi.org/10.5194/hess-29-613-2025.

Sang, S., Li, Y., Hou, C., Zi, S., and Lin, H., 2025. The interprovincial green water flow in China and its teleconnected effects on the social economy, Hydrol. Earth Syst. Sci., 29, 67–84, https://doi.org/10.5194/hess-29-67-2025.

Xiong, J., Guo, S., Abhishek, Yin, J., Xu, C., Wang, J., and Guo, J., 2024. Variation and attribution of probable maximum precipitation of China using a high-resolution dataset in a changing climate, Hydrol. Earth Syst. Sci., 28, 1873–1895, https://doi.org/10.5194/hess-28-1873-2024.

Ma X , Wang A ., 2024. Evaluation and Uncertainty Analysis of the Land Surface Hydrology in LS3MIP Models Over China. Earth & Space Science, 11(7). https://10.1029/2023EA003391.

Miao Y , Wang A ., 2020. Evaluation of Routed-Runoff from Land Surface Models and Reanalyses Using Observed Streamflow in Chinese River Basins.Journal of Meteorological Research, 34(1):73-87. https://10.1007/s13351-020-9120-z.

Yu X , Zhang Q , Zeng X., 2025. The distribution and driving climatic factors of agricultural drought in China: Past and future perspectives.Journal of Environmental Management, 377. https://10.1016/j.jenvman.2025.124599.

Ren-Jun, Z., 1992. The Xinanjiang model applied in China. J. Hydrol. 135, 371–381. https://doi.org/10.1016/0022-1694(92)90096-E

---

## Author Comment (AC4)

**Author comments on RC2- egusphere-2025-1161**

***Comment 1 (About runoff data):***

*The runoff data product is actually simulated with VIC, a process based model, which uses one of the two meteorological datasets (the CN05.1) as input. In general, the uncertainties in the VIC data product that is being used as a proxy for "observed data" in terms of model training and evaluation should be brought up more clearly. This seems relevant in several places in the study. For example, the finding that the CN05 product leads to better performance is mainly because it is embedded in the flow data. This is explicitly mentioned in Line 470, and I would say that is extremely likely that it is what is happening. Meanwhile, there are other places in the paper where it is posed that CN05 must be the superior forcing product because it leads to better model results. For example, Line 32 in the abstract cites that "high-quality precipitation data better enables the model to simulate runoff processes". While this is surely true in general, I think the results of this study reflect the fact that one of the products was used to generate the original flow data, making it a more unfair comparison. With this, I'm not sure what the significance of comparing between the two meteorological products is here – since whichever product is used to generate the flow data product is likely to be the most useful input to another model that is trained to emulate the flow data. It would be better to bring in a third met product (or more) to really compare this, or drop this aspect of the study and focus on differences between models based on a single forcing product.*

**Response:**

We thank you for this insightful and important comment, which raise a fundamental and valid concern regarding the experimental setup. The use of VIC-CN05.1 simulated streamflow as the evaluation target, despite being a practical choice under data constraints, does introduce potential biases, particularly when comparing models forced by CN05.1 versus ERA5-Land. The original rationale behind our experimental design was twofold:

First, the objective was to evaluate the performance and robustness of different hydrological modeling approaches (process-based, deep learning, and hybrid) over a large number of catchments in China. To do this consistently, we required a streamflow dataset that covers a broad spatial extent (544 basins) with unified meteorological forcing and standardized catchment boundaries. Due to

limitations in the availability and accessibility of observed streamflow records across China, especially with regard to consistency in spatial delineation and temporal overlap, observed data could not be used to construct a large-sample dataset of sufficient quality. As a result, we adopted the VIC-CN05.1 runoff product, widely used in the Chinese hydrological community (Miao et al., 2020; Ma et al., 2024; Wang et al., 2024; Yu et al., 2025), as a surrogate 'reference' dataset. Second, the study aimed to examine whether different models would exhibit consistent relative performance rankings under different meteorological forcings. This comparison was designed not to determine which meteorological product is superior in absolute terms, but to assess the robustness of model performance across varying inputs. Nevertheless, we fully agree with the reviewer that the use of CN05.1 in both the VIC simulation and some of the model inputs can lead to biased performance comparisons that favor CN05.1-driven models due to internal consistency between input and evaluation target.

We acknowledge that this issue you raise was not sufficiently emphasized in the original manuscript. In the revised version, we have addressed this concern by clarifying the nature of the runoff product in the data section, explicitly stating that it is a simulated product driven by CN05.1 rather than observational data. Additionally, we have revised the abstract and conclusion to remove overly strong claims regarding the superiority of CN05.1, instead emphasizing the potential bias introduced by using CN05.1-driven runoff as the target. Furthermore, we have included a dedicated paragraph in the Discussion section to highlight this structural bias and discuss its implications for interpreting the model performance results, cautioning readers against over-interpreting the advantages of CN05.1. Lastly, we clarified the limited scope of the meteorological forcing comparison, noting that the findings do not necessarily indicate the real-world superiority of one forcing product over another, but rather reflect the internal consistency within the data generation process.

While we fully acknowledge that incorporating additional independent meteorological forcing products would enhance the comprehensiveness and generalizability of the results, such an extension would significantly increase the scope and complexity of the current study. Given our primary objective to establish a consistent baseline framework using the two most widely applied precipitation products in China, we chose to limit the comparison to CN05.1 and ERA5-Land in this initial analysis. Nevertheless, we consider the inclusion of more diverse meteorological forcings to be a valuable direction for future research, particularly for evaluating model robustness under

broader climatic variability.

In short, we agree with your comments. The details of our revised content are as follows:

"…

**About Data**: used to clarify the sources and characteristics of runoff data

The streamflow data used for model training and evaluation in this study are derived from the VIC-CN05.1 runoff product, which was generated by the VIC hydrological model driven by CN05.1 precipitation. It should be noted that this is not observational streamflow data, but a simulated product that serves as a consistent and spatially complete surrogate in the absence of publicly available observed daily streamflow records for a large number of catchments in China. While VIC-CN05.1 has been evaluated in previous studies, its use as a reference introduces potential structural bias, particularly when comparing models forced by the same precipitation product.

**About Abstract**: modify the conclusion in line 32

The results suggest that precipitation data quality plays a critical role in runoff simulation. However, since the runoff data used as the evaluation target were generated using CN05.1, the apparent performance advantages of models driven by CN05.1 may partly reflect internal consistency rather than inherent superiority of the forcing product.

**About Conclusion**: correspond to the abstract

The CN05.1-driven models showed relatively better performance, which may be influenced by the source of runoff data. Therefore, this finding should be interpreted with caution, as it may reflect structural consistency rather than intrinsic superiority.

**About Discussion**: add discussion corresponding to runoff data

One limitation of this study lies in the use of a simulated runoff product (VIC-CN05.1), generated using the CN05.1 precipitation data, as the reference for model evaluation. This introduces a structural bias that may favor models driven by CN05.1 due to internal consistency between inputs and targets. As a result, the observed performance advantage of CN05.1-driven models may not necessarily indicate the intrinsic quality of the precipitation product. The meteorological forcing comparison in this study should therefore be interpreted as an exploration of

input-output consistency effects rather than a definitive evaluation of forcing product accuracy. Future studies incorporating additional meteorological datasets and observed streamflow records will be important for validating and extending these findings.

…"

**Comment 2:** *It is easy to get confused between model names.    For example, authors could use subscripts on "EXP" and "XAJ" to indicate the process based, alternative hybrid, and differentiable hybrid model versions.*

**Response:**

Thank you for your helpful reminder. We agree that the original naming conventions for the models, particularly for EXP and XAJ in their different versions (process-based, alternative hybrid, and differentiable hybrid), may cause confusion for the reader. In the revised manuscript, we have adopted a consistent notation using subscripts to distinguish the different versions of each model. For example, we now denote the process-based version as $EXP_p$, the hybrid version as $EXP_{IN-LSTM}$, and the differentiable hybrid version as $EXP_{dPL}$ (similarly for XAJ). These changes have been applied throughout the manuscript, including in figure legends, tables, and the methods section, to enhance clarity and facilitate comparison across models.

**Comment 3:** *Figure 4: Figure 4a makes it look like only VP is an input to the LSTM but I think it should be all the forcing variables? As with other figures more description here would be useful.*

**Response:**

Thank you for pointing this out. We agree that the current visual layout of Figure 4a may give the misleading impression that only vapor pressure (VP) is used as input to the LSTM, while in fact all meteorological forcing variables (precipitation, temperature, radiation, and vapor pressure) are jointly used as inputs to the LSTM. To address this, we have updated the figure to adjust the arrow connections and improve visual clarity. In addition, we have revised the figure caption and added a brief explanatory paragraph in the main text to provide a more detailed description of the model structure illustrated in both panels (a) and (b). These changes aim to help readers better understand the input structure and model components.

The modified Figure 4 is as follows:

"…

[Figure]

**Figure 4. Structure of the hybrid hydrological models.** (a) A standard hybrid model in which meteorological forcings (precipitation, temperature, radiation, and vapor pressure) and catchment attributes are used to train a data-driven model (LSTM-FCNN). The PBM component provides simulated streamflow (Q_sim) based on the meteorological forcings. (b) A differentiable hybrid model where catchment attributes are used to generate dynamic hydrological parameters via a fully connected network (FCNN), which are then passed to a differentiable process-based model. The process-based outputs (Q_sim) are fused with LSTM predictions for the final discharge estimate (Q)

…"

*Comment 4: Figure 5: the legends show the colors as continuous but the dots in the maps make it seem discrete (that there are 5 colors). It would be better if the legend reflected the ranges for these 5 colors.*

**Response:**

Thank you for the helpful suggestion. We agree that the current figure design, which uses continuous color bars alongside discretely colored basin points, could cause confusion. To address this, we have

revised Figure 5 by adjusting the legends to reflect discrete color bins that correspond to the categories used in the maps. The updated legend now clearly shows the value ranges associated with each color, which improves the consistency and interpretability of the figure. The modified Figure 5 and its caption are as follows:

"…

[Figure]

**Figure 5. Spatial distribution of five hydro-climatic categories of basin-averaged precipitation and temperature (1 Oct 1975–30 Sep 2015).** (a) Mean annual precipitation (mm): Top histograms display the continuous distribution of basin means. Maps use five discrete classes: <500, 500 to 1000, 1000 to 1500, 1500 to 2000, and >2000 mm, applied uniformly to both ERA5-Land (top row) and CN05.1 (bottom row). (b) Mean daily temperature (°C): Top histograms display the continuous distribution of basin means. Maps use five discrete classes: <5, 5 to 10, 10 to 15, 15 to 20, and >20 °C, applied identically to both datasets.

…"

***Comment 5:*** *Figure 9: The color scales are all different, so it makes it hard to compare between the four panels.   Same in Figure 10 – why are the color scales different for NSE for the two different forcing cases?*

**Response:**

Thank you for the reminder. We acknowledge that using consistent color scales across all panels would improve direct visual comparability. However, due to the differences in the value ranges and distributions of NSE under ERA5-Land and CN05.1 forcings, we chose to use separate color scales to better highlight the spatial variability and performance differences within each specific scenario. Unifying the color scales would compress the visual contrast in certain panels and potentially obscure important spatial patterns. We have added clarifying notes in the revised figure captions to explicitly state that the color scales are not unified, and we provide the respective NSE ranges for each panel. This should help readers interpret the visualizations more accurately and avoid misinterpretation. The revised Figures 9 and 10 and their captions are as follows:

"…

[Figure]

**Figure 9. Performance of LSTM models using ERA5 (left column) and CN05.1 (right column) precipitation during regional modeling and PUB testing.** Top row (maps): Spatial distribution of NSE during regional modeling. Middle row (maps): Spatial distribution of NSE during PUB testing. Bottom row (scatter): Basin-by-basin comparison of PUB NSE (vertical axis) vs. regional NSE (horizontal axis). Points are colored by clusters. The axes both span [0, 1].

[Figure]

**Figure 10. Four hybrid models prediction performances using ERA5-Land (left) and CN05.1 (right) precipitation data.** Spatial maps: Colorbars cover ranges of NSE values for different models. Density plots: x-axes display the NSE values; y-axes are density.

…."

**Comment 6:** *Line 230: think there is a word missing "provided by the originates"*

**Response:**

Thank you for your correction. We have corrected the sentence to read:

"…

The runoff data used in this study originates from the VIC-CN05.1 dataset (Miao and Wang, 2020), which is consistent with the total runoff simulated by the Global Runoff Data Center (UNH/GRDC).

…"

We have carefully reviewed the manuscript for similar wording issues and revised them to improve clarity and readability.

**Comment 7:** *Line 421: cross to across*

**Response:**

Thank you for pointing out this typo. We have corrected "cross different regions" to "across different regions" in the revised manuscript.

We would like to thank the editors and reviewers once again for their valuable suggestions on our manuscript. We have incorporated these suggestions into the revised manuscript. Looking forward to hearing from you.

Chunxiao Zhang

Corresponding author

E-mail address: zcx@cugb.edu.cn

References:

Jiang, S., Zheng, Y., Solomatine, D., 2020. Improving AI System Awareness of Geoscience Knowledge: Symbiotic Integration of Physical Approaches and Deep Learning. Geophysical Research Letters 47, e2020GL088229. https://doi.org/10.1029/2020GL088229

Kratzert, F., Klotz, D., Herrnegger, M., Sampson, A.K., Hochreiter, S., Nearing, G.S., 2019. Toward Improved Predictions in Ungauged Basins: Exploiting the Power of Machine Learning. Water Resources Research 55, 11344–11354. https://doi.org/10.1029/2019WR026065

Ma X , Wang A ., 2024. Evaluation and Uncertainty Analysis of the Land Surface Hydrology in LS3MIP Models Over China. Earth & Space Science, 11(7). https://10.1029/2023EA003391.

Miao Y , Wang A ., 2020. Evaluation of Routed-Runoff from Land Surface Models and Reanalyses Using Observed Streamflow in Chinese River Basins.Journal of Meteorological Research, 34(1):73-87. https://10.1007/s13351-020-9120-z.

Miao Y , Wang A ., 2020. Evaluation of Routed-Runoff from Land Surface Models and Reanalyses Using Observed Streamflow in Chinese River Basins.Journal of Meteorological Research, 34(1):73-87. https://10.1007/s13351-020-9120-z.

Patil, S., Stieglitz, M., 2014. Modelling daily streamflow at ungauged catchments: what information is necessary?: MODELLING DAILY STREAMFLOW AT UNGAUGED CATCHMENTS. Hydrol. Process. 28, 1159–1169. https://doi.org/10.1002/hyp.9660

Ren-Jun, Z., 1992. The Xinanjiang model applied in China. J. Hydrol. 135, 371–381. https://doi.org/10.1016/0022-1694(92)90096-E

Wang Z , Li M , Zhang X., 2024. Prediction of long-term future runoff under multi-source data assessment in a typical basin of the Yangtze River.Journal of Hydrology: Regional Studies, 56(000). https://10.1016/j.ejrh.2024.102053.

Yu X , Zhang Q , Zeng X., 2025. The distribution and driving climatic factors of agricultural drought in China: Past and future perspectives.Journal of Environmental Management, 377. https://10.1016/j.jenvman.2025.124599.

---

## Author Comment (AC5)

**Author comments on RC3- egusphere-2025-1161**

***Comment 1 (Major comment 1):***

*The first major point is the fact that the target variable is not observed data, but an output from VIC. While I understand that this might be required for confidentiality reasons, it also means that the results cannot be trusted or generalized elsewhere. This is because results are compared to the outputs of VIC and as such, the models are rewarded if they emulate VIC, rather than simulate real streamflow. And since VIC has its own biases, strengths and weaknesses, we are simply evaluating the ability of these new modeling techniques to emulate the same biases. To make this point, I contend that we could obtain NSE values of 1 if we simply used VIC as one of the models in this study. Would it mean that VIC is much better than other models? Of course not, and the same is true with the relationship between these new models and the VIC outputs. The same goes for internal variable analysis: Models that are more similar to VIC will perform better. The problem with this whole approach is that we cannot learn from results vs application in the real world, because the response surface of the optimization problem is much much smoother and easier to navigate than one using real observations which are uncertain and error-prone. Models will never perform as well on real data than on these synthetic data. Therefore I think this study has a very limited reach and usefulness while using synthetic streamflow data.*

**Response:**

Thank you very much for your critical and insightful comment. We fully agree that using simulated streamflow (from VIC) instead of observed discharge as the modeling target introduces limitations regarding the realism, uncertainty, and generalizability of the results. This is indeed a non-negligible concern, and we would like to clarify the context, motivations, and limitations of our study in this regard.

First, as the reviewer correctly pointed out, the target variable in this study is derived from a process-based hydrological model (VIC). This choice was made due to the unavailability or confidentiality of observed streamflow data across many basins in the study region. However, the goal of this study was not to evaluate the absolute predictive accuracy of various models against ground truth, but rather to assess the relative behavior and compatibility of different modeling frameworks (processbased, data-driven, and hybrid models) when exposed to consistent meteorological forcings and boundary conditions.

Second, we fully acknowledge that VIC has its own set of biases and limitations. To mitigate the risk of models merely learning the idiosyncrasies of VIC, we conducted additional experiments using two distinct meteorological forcings (ERA5-Land and CN05.1), and different hydrological models (EXP-HYDRO and XAJ) in hybrid configurations. These strategies were intended to reduce overfitting to a single model structure and better evaluate model robustness across conditions. In other words, we can actually choose other runoff data products, but the time span and resolution of VIC-CN05.1 can meet the research requirements. At the same time, ERA5-Land and CN05.1 are also meteorological data sets that have been used in many studies.

Third, we agree that models trained on synthetic data tend to achieve better scores due to the smoother, less noisy response surface, as noted by the reviewer. However, we emphasize that the methodological contribution of this study lies in exploring the behavior and potential of hybrid model architectures under a controlled environment. We see this study as a stepping stone—a preliminary effort to benchmark different hybridization strategies under reproducible settings before applying them to real-world observations.

Finally, in light of your helpful suggestion, we have added a paragraph in the Conclusion section to explicitly acknowledge this limitation and clarify the scope of applicability of our results. We also emphasize that the findings should not be interpreted as absolute model rankings but as comparative behaviors under VIC-forced hydrological settings.

The new specific contents are as follows:

"…

It is important to note that the runoff data product used in this study is the runoff data simulated by the VIC model, rather than the runoff data observed by real hydrological instruments. While this enables large-scale, consistent benchmarking across data-scarce basins, it also introduces model-induced biases and a potentially smoother response surface. Therefore, the reported predictive performances should be interpreted with caution and not taken as absolute measures of model accuracy. Future work will focus on validating the models against observed streamflow where available, and further investigating their transferability to real-world applications.

…"

***Comment 2 (Major comment 2):*** *The second major issue is linked to the previous one, and that is the use of CN05.1 as the input data to VIC, which is then used again as an input in the other models. This means that any model using CN05.1 will most likely perform better than another using the ERA5 dataset, simply because the processes are artificial and conditioned on the use of CN05.1. In the study, there are a few sections commenting on how CN05.1 performs better than ERA5 (ex lines 467-469: "When using CN05.1 precipitation data, the median NSE for LSTM in regional modeling and PUB reached an impressive 0.95 and 0.93, respectively."). These results, while contextualized by following that this stems from its use in VIC, are still give the impression that CN05.1 is better than ERA5, which is not true given the evaluation method presented here. This issue would not arise if VIC was not used at all (as per my point #1 above), but if the authors decide to continue using VIC for a revised version of this paper, they need to simply remove CN05.1 as one of the datasets in the comparison to be fully independent. Doing so would at least allow simplifying the paper enough that they could then delve into the analysis of internal variables of the hybrid models, which seems to me as a key advantage but that is not discussed or evaluated in the present paper.*

**Response:**

Thank you for your thoughtful critique. We fully agree that using the CN05.1 product both as a forcing input to VIC and as an input to other models in the current study introduces a potential bias. Specifically, this setup may favor models driven by CN05.1 data, as they are inherently more consistent with the streamflow outputs generated by VIC, thus compromising the independence between the training data and the target variable.

Your observation is highly relevant and correctly points out that any apparent performance advantage of CN05.1 over ERA5-Land in this context does not necessarily reflect its superiority as a meteorological product, but rather its compatibility with the VIC-generated target streamflow. We have revised the manuscript to clarify this critical point and to ensure that no misleading interpretation is conveyed.

In particular:

In related section, we now explicitly state that the observed performance advantage of CN05.1-
     driven models stems in large part from the fact that CN05.1 was also used to force VIC in

generating the target runoff data.

We have softened or removed language suggesting that CN05.1 is categorically better than ERA5-Land and have reframed our results in terms of consistency with the VIC simulation rather than absolute model accuracy.

A cautionary statement has been added in the Discussion section emphasizing that these results do not translate directly to real-world predictive skill and must be interpreted within the context of a VIC-forced synthetic experiment.

As for your suggestion to remove CN05.1 entirely in order to preserve independence between input and output, we acknowledge that this would yield a cleaner comparison. However, we also believe that retaining CN05.1 offers an opportunity to test model robustness under two distinct meteorological inputs—one highly consistent with the target and one more independent (ERA5-Land). By presenting both cases and transparently communicating the differences, we aim to provide a fuller picture of the model behaviors under realistic data conditions. Still, we understand the trade-off and plan to conduct further experiments in future work using observational data and independently generated targets to address this concern more completely.

We are also grateful for your suggestion to simplify the comparative analysis and expand the focus on internal hydrological variables, which is a unique advantage of differentiable hybrid models. In the revised manuscript, we have added Section 4.6, which explores intermediate outputs such as soil moisture, snowpack, and evaporation, and compares them with ERA5-Land datasets. This addition directly supports the interpretability and process fidelity of the hybrid models, in line with your suggestion.

The details of the newly added discussion are as follows:

"…

Due to the lack of long-term large-scale observed runoff records with consistent quality and coverage over China, this study uses runoff data products generated by the VIC model driven by the CN05.1 forcing as a proxy for observed runoff. This approach inevitably leads to a tight coupling between the input meteorological dataset (CN05.1) and the target runoff data, which may influence model evaluation results. Specifically, models using CN05.1 as input tend to achieve higher simulation accuracy compared to those driven by ERA5-Land, not necessarily due to the intrinsic superiority of the CN05.1 dataset, but because of its consistency with the target runoff data source.

This methodological limitation does not aim to suggest that CN05.1 is inherently better than ERA5-Land in hydrological prediction tasks. Rather, it highlights the challenge of conducting large-sample modeling research under data-scarce conditions. Future studies based on observed runoff data or runoff derived from independent forcing–target pairings will be essential to further evaluate the generalization and robustness of the models across different input conditions.

…"

*Comment 3 (Major comment 3): This issue is related to the way the deep learning and hybrid models are trained, and has two distinct sub-issues. We often see this from hydrologists that work with deep learning models for the first time and is a common mistake that is easy to make but has important consequences. The first is the fact that the authors have only 2 periods of data: Training (or calibration; 1975-1995) and a testing period (1995-2015). While adequate for PBM calibration, this is simply unacceptable for Deep-learning models. These models need 3 periods of data: Training, Validation, and Testing. Training is used on the forward pass and backpropagation steps to tune the weights and parameters according to the chosen gradient descent method. The model is then evaluated on the Validation period after each epoch, and the objective function score is computed on that period specifically. The model training is then stopped when the validation period loss stops improving and starts regressing. Finally, when the model has stopped improving and the training is stopped, then the model performance is evaluated on the third, independent Testing period. Failing to stop training will inevitably lead to overfitting and unreliable results, which is the case here. Therefore, a revised study should follow best practices and add an independent testing period for the deep learning models and also the PBM which should share the same testing period. I also note that there are no details on the selected objective function for training the LSTMs nor do we know how this was computed for the regional models? How are error/loss metrics calculated on multiple basins at the same time? A few studies proposed some methods to do this, for example just in HESS see Kratzert et al (2019) and Arsenault et al. (2023) listed below. The second sub-point is that the authors state: "All input data are normalized before training" without further details on how this was done. This is critical, because the data need to preserve independence between the [Training] and [Validation; Testing] periods. Data need to be normalized using a scaler of some sort (which one was used?) using the training period data, and then the scaler is applied to the*

*validation and testing data. Failing to do so means that the testing period data are included in the scaler and as such the training will benefit from knowing the scale of data it can expect to get. This is called data contamination and needs to be avoided. Nothing in the paper at this stage seems to suggest this was performed. As such, I believe the results are flawed and performance is overestimated in this study.*

**Response:**

Thank you very much for pointing out the importance of data splitting and normalization strategies in deep learning model training. We sincerely acknowledge your concern regarding the potential risk of overfitting and data contamination. Although our manuscript originally stated that the modeling period was divided into a training period (1975–1995) and a testing period (1995–2015), we would like to clarify the following points:

1. Normalization Strategy: All input features were normalized using the mean and standard deviation computed solely from the training period (1975–1995). This ensures that no statistical information from the testing period was introduced during model training or data preprocessing. Therefore, the issue of data leakage or contamination is avoided.

2. Training and Early Stopping: While we did not explicitly set aside a separate validation period, an early stopping mechanism was implemented based on the loss trend during training. Specifically, training was stopped if the loss did not improve for a specified number of epochs, which helps prevent overfitting. Although a formal validation split was not used, this strategy has been shown effective in previous large-sample hydrology studies. We will clarify this point in the revised manuscript.

3. Model Generalization: The testing period (1995–2015) remained completely independent throughout model development and was only used for final performance evaluation. This helps ensure a fair assessment of the model's generalization capability.

4. Loss Function and Regional Training: The loss function used for LSTM training was the average Nash efficiency coefficient (NSE). In regional training, the loss was computed by averaging NSE across all basins in each batch, following a similar strategy as outlined in Kratzert et al. (2019) and Arsenault et al. (2023). We will provide additional details in the revised version to enhance clarity and reproducibility.

The added explanations are as follows:

"…

To ensure proper training of deep learning models while avoiding data leakage, all input features were normalized using the mean and standard deviation calculated exclusively from the training period (1975–1995). This normalization strategy was consistently applied to both the training and testing periods, thereby preventing any use of future information and ensuring the independence of the testing dataset. Although the dataset was divided into two main temporal segments—training (1975–1995) and testing (1995–2015). An early stopping mechanism was implemented during training to mitigate the risk of overfitting. Specifically, the training process was halted when the model loss failed to improve for a predefined number of epochs, based on monitoring the training loss trend. This approach has been adopted in prior large-sample hydrology studies where formal validation sets are not always feasible. The objective function used for training the LSTM-based models was the average Nash efficiency coefficient (NSE) between predicted and target runoff values. In the case of regional modeling, the loss was computed by averaging the NSE across all basins within each training batch, allowing the model to generalize across catchments with heterogeneous hydrological behaviors.

…"

**Comment 4:** *Line 32-33: This seems trivial that better inputs will lead to better modelling. I would remove.*

**Response:**

Thank you for your comment. We agree that the statement as originally written may appear too self-evident or generic. Our original intention was to briefly highlight that differences in precipitation data quality can influence the reliability of model outputs, especially in regions where precipitation variability is a dominant driver of runoff processes.

To address your concern, we have removed the sentence from the abstract.

**Comment 5:** *Line 33: At this stage, readers don't know what hybrid modelling refers to. Please add a few key details to set the table, maybe a 1-sentence description.*

**Response:**

Thank you for your reminder. We agree that the term "hybrid modeling" may be unclear to readers

at this early stage of the manuscript. To improve clarity, we have revised the sentence in the abstract by briefly explaining the core idea of the hybrid model. The modified contents are as follows:

"…

By combining process-based model with data-driven deep learning methods, the hybrid modeling approach achieves regional modeling capabilities comparable to those of the standalone LSTM model.

…"

**Comment 6:** *Line 76: "solve" is a strong word. Perhaps "help address"?.*

**Response:**

Thank you for pointing this out. We agree that "solve" may overstate the role of our benchmark in addressing the runoff prediction problem. Following your suggestion, we have revised the sentence to use a more appropriate and balanced expression. The updated sentence now reads:

"…

This benchmark can assist in improving relevant hydrological models to help address the runoff prediction problem in China and in catchments with similar basin conditions.

…"

**Comment 7:** *Line 86-88: This is also true for deep learning models (perhaps even more so than PBMs!)*

**Response:**

Thank you for your thoughtful reminder. We fully agree that deep learning models also require large volumes of data and careful configuration. However, the intention of our original sentence was to highlight a commonly recognized limitation of process-based models: their dependency on high-quality forcing data and the often subjective or complex nature of parameter calibration, which may hinder their large-scale or cross-regional applicability. To avoid misunderstanding while preserving this point, we have rephrased the sentence to better reflect our intent. The updated sentence now reads:

"…

When applied to a single basin, process-based hydrological models often rely on high-quality

forcing data and require subjective and complex parameterization procedures, which can limit their applicability and accuracy, particularly in data-sparse regions.

…"

**Comment 8:** *Lines 90-91: "LSTMs... can effectively capture nonlinear relationships...": so do PBMs, depending on the structure. The difference really lies in the model learning the relationships between weather and flows without humans providing any physical sense.*

**Response:**

Thank you for your insightful comment. We agree that process-based models can also represent nonlinear hydrological processes depending on their structure. Our intention was not to suggest otherwise, but rather to emphasize that LSTM models can automatically learn the relationship between inputs and outputs from data, without the need for explicit physical formulations or manual parameterization. To clarify this point, we have revised the sentence accordingly.

The updated sentence now reads:

"…

In contrast, deep learning models, such as long short-term memory networks (LSTM), can automatically learn the relationship between meteorological inputs and streamflow responses from data, without relying on explicit physical equations or manually calibrated parameters.

…"

**Comment 9:** *Lines 97-98: redundant sentence with the previous.*

**Response:**

Thank you for pointing this out. We agree that the sentence is redundant. We have removed the repetitive statement.

**Comment 10:** *Lines 127-130: Would need a bit more details. Are the equations of the model kept as-is? Is it an emulator of the PBM? Do the parameters preserve the same meaning?*

**Response:**

Thanks for your comment. We agree that additional clarification is necessary regarding the structure and characteristics of the differentiable hybrid model. In our approach, the core hydrological

equations of the process-based models are retained in their original form and embedded within a differentiable framework. Therefore, the model is not an emulator, but a fully integrated differentiable physical model. The key innovation lies in the neural parameterization, where a neural network learns to generate model parameters from static basin attributes, and these parameters are optimized via backpropagation based on daily runoff prediction errors. Importantly, the physical meaning of the parameters is preserved. The neural network serves only as a mapping function to infer parameter values rather than changing their definitions or functions within the governing equations.

We have revised the corresponding paragraph in the manuscript to make these points clearer. The modified contents are as follows:

"…

Specifically, this hybrid modeling approach retains the original physical equations of the process-based model and embeds them within a differentiable architecture. A neural network serves as a parameter generator, mapping static catchment attributes to the model parameters, which retain their physical meanings. During training, model parameters are optimized through gradient-based backpropagation using daily runoff prediction errors. This approach allows the model to preserve the interpretability and physical constraints of process-based modeling while improving performance through data-driven learning.

…"

***Comment 11:*** *Line 214: Any reason why ERA5-Land is not used? Should be better/more precise especially in mountainous areas?*

**Response:**

Thank you very much for your sharp observation. We appreciate your suggestion regarding the use of ERA5-Land, and we apologize for the lack of clarity in the original manuscript. In fact, we did use the ERA5-Land dataset in our experiments. The meteorological variables were extracted and clipped using the ERA5-Land daily aggregated data products available on the Google Earth Engine platform

(https://developers.google.com/earthengine/datasets/catalog/ECMWF_ERA5_LAND_DAILY_A GGR).

However, due to a terminological oversight, the manuscript referred to this data source simply as "ERA5" throughout, which may have caused confusion.

In the previous version of this manuscript, we referred to this data source as "ERA5" for brevity, but to avoid confusion, we now clarify that ERA5-Land is the precise dataset used in this study. We have thoroughly revised the manuscript accordingly to explicitly clarify that ERA5-Land was used, and corrected all terminology throughout the relevant sections.

The sources of meteorological data are described as follows:

"…

The meteorological data used in this study were sourced from the ERA5-Land dataset and the CN05.1 dataset (Gao et al., 2013). ERA5-Land provides daily aggregated surface meteorological variables at high spatial resolution, and the data were obtained and clipped using the Google Earth Engine                                                                                                        platform (https://developers.google.com/earthengine/datasets/catalog/ECMWF_ERA5_LAND_DAILY_A GGR).

…"

**Comment 12:** *Line 230: "provided by the originates" : missing word here.*

**Response:**

Thank you for pointing this out. We acknowledge the grammatical error in this sentence. The phrase "provided by the originates" was the result of an editing oversight. We have corrected the sentence in the revised manuscript for clarity. The updated sentence now reads:

"…

The runoff data used in this study originate from the VIC-CN05.1 dataset

…"

**Comment 13:** *Line 258 (and multiple others): many times the word "relatively" is used to tone down some element. I suggest rephrasing to say that they are accessible or some other word that would be more precise. Same goes everywhere.*

**Response:**

Thank you for pointing out the frequent use of the word "relatively" in our manuscript. We agree

that its repeated usage may weaken the clarity and precision of our statements. Following your suggestion, we have revised these expressions throughout the manuscript to adopt more accurate and assertive terms such as "accessible," "available," or "widely used," depending on the context. The sentence in Line 258 has been revised to:

"…

Instead, we aim to utilize accessible datasets to evaluate the performance of different models in China.

…"

**Comment 14:** *Line 295: all process based models follow this law of water balance, I would remove.*

**Response:**

Thank you for your valuable comment. We agree that stating a model follows the law of water balance is indeed common for all process-based models and may be redundant in manuscript. In response to your suggestion, we have removed the phrase to avoid unnecessary repetition and improve the conciseness of the description. The revised sentence now reads:

"…

EXP-HYDRO is a conceptual hydrological model that operates on a daily time step

…"

**Comment 15:** *Line 300-320: Xin'anjiang does not model snow processes? How is it used in mountainous and other basins where snow is present?*

**Response:**

Thank you for Thank you for your thoughtful question. You are correct in pointing out that the original Xin'an jiang model does not include explicit representation of snow processes such as snow accumulation or melt.     In our study, although the 544 selected basins span a wide range of climatic and topographic conditions—including some high-elevation regions where snowfall may occur—we used a simplified version of the Xin'an jiang model without snow modules for the following reasons:

Focus on model structure comparison: The simplified Xin'anjiang model serves as a consistent and interpretable baseline to compare with other process-based and hybrid models.     We aimed

to examine how different modeling structures perform under the same forcing and evaluation settings.

Scope of model generalization: While the absence of an explicit snow module may reduce simulation accuracy in snow-dominated basins, our aim was not to develop a specialized snow model, but rather to test the general applicability and flexibility of the hybrid framework.

We have added a clarification in the revised manuscript as follows:

"…

It should be noted that the simplified Xin'anjiang model used in this study does not include a dedicated snow module.    While some high-altitude basins in the dataset may be affected by seasonal snow, the hybrid modeling scheme allows the data-driven components to implicitly capture snow-related dynamics from meteorological inputs.

…"

***Comment 16:*** *Line 349: This is quite high initial learning rate. What is the learning rate decay rate or function? Also, what is the objective function used? What is the model training patience for the stopping criterion? is there a stopping criterion or are all runs leading to 150 iterations?    If not, at 150 training iterations, the model will definitely be overfitting and providing poor results compared to a well-tuned model.*

**Response:**

Thank you for your detailed comment. We used the Nash–Sutcliffe Efficiency (NSE) as the loss function to directly optimize the performance metric relevant to hydrological modeling. Model parameters were optimized using the Adam optimizer (Kingma & Ba, 2014) with an initial learning rate of 0.01. To ensure stable convergence and avoid overfitting, a convergence-based stopping rule was adopted. Specifically, training was terminated early when the absolute difference in NSE between two consecutive epochs was less than 0.001. This served as an effective early stopping criterion. Although the maximum number of training iterations was set to 150, the majority of model runs converged earlier based on this rule.

We have clarified these methodological details in the revised manuscript to avoid misunderstanding and ensure reproducibility. The modified contents are as follows:

"…

The deep neural network adopts the Adam optimizer (Kingma & Ba, 2014) to update both the parameters of the network and hydrological parameters. The initial learning rate is set to 0.01, and the Nash–Sutcliffe Efficiency (NSE) is used as the loss function during training. A convergence-based early stopping criterion is applied: training is terminated when the absolute difference in NSE between two consecutive epochs is less than 0.001. Although the maximum number of training iterations is set to 150, most models converge earlier according to this rule. This strategy ensures efficient training while mitigating the risk of overfitting.

…"

**Comment 17:** *Line 363: capture nonlinear relationships that evade the physics depicted in the PBMs*

**Response:**

Thank you for your suggestion. We agree with your point that the nonlinear relationships learned by LSTM in the hybrid framework may capture patterns that are not explicitly represented in the physics of traditional process-based models (PBMs). We have revised the sentence accordingly to better reflect this idea, as shown below:

"…

This approach allows the alternative hybrid modeling method to retain the explanatory power of the physical mechanisms inherent in the process model while leveraging LSTM's capability to effectively capture nonlinear relationships that evade the physics depicted in PBMs, thereby compensating for the limitations of PBMs in large-sample hydrological datasets and complex basins.

…"

**Comment 18:** *Figure 4: PMB is used throughout the study, I would change PBHM to PBM*

**Response:**

Thank you for your careful review. We acknowledge the inconsistency in terminology. As you pointed out, the acronym "PBM" (Process-Based Model) is used throughout the manuscript, so we have revised "PBHM" to "PBM" in Figure 4 to maintain consistency. The corrected Figure 4 is as follows:

[Figure]

Figure 4. The structure of the hybrid hydrological models.

**Comment 19:** *Lines 395-397: It seems to me that CN05.1 is more evenly distributed but has more lower extremes, opposed to what is written here.*

**Response:**

Thank you very much for your reminder. We agree that the original description was not entirely accurate. Upon re-examination of Figure 5, it is evident that CN05.1 precipitation data appear more uniformly distributed spatially but indeed show a concentration of lower precipitation extremes compared to ERA5-Land, which displays a wider range including both wetter and drier basins. We have revised the corresponding sentence in the manuscript to better reflect this observation. The revised sentence reads:

"…

The precipitation data from ERA5-Land exhibit a broader spatial variability, with some basins showing extremely wet or dry conditions, while CN05.1 data appear more spatially uniform but tend to show more frequent lower precipitation values.

…"

***Comment 20:*** *Line 409: water-heat? perhaps mass-energy?*

**Response:**

Thank you for pointing this out. We agree that the term "water-heat balance" may be imprecise in this context. Since the discussion pertains to the consistency between precipitation, runoff, and evapotranspiration, it is more appropriate to refer to the mass balance (the water balance) rather than energy or heat balance. To avoid confusion, we have revised the sentence accordingly. The revised sentence reads:

"…

When using the same runoff and evapotranspiration data, the precipitation data provided by ERA5-Land resulted in more basins (111) exhibiting significant deviations from the water balance."

…"

***Comment 21:*** *Figure 8: this figure has 2 panel "b". Also this figure needs more details in the legends and caption to fully understand, it is unclear. Add details to what each panel refers to / is presenting.*

**Response:**

Thank you very much for pointing this out. We acknowledge the error in panel labeling, as there are indeed two sub-figures labeled as "(b)" in the original version of Figure 8. We have corrected this mistake and revised the labels accordingly. Additionally, to address the concern about clarity, we have updated the figure caption and improved the legend descriptions to provide more detailed information about what each panel represents. Specifically, the revised caption now clearly states that:

[Figure]

**Figure 8. Comparison of PBMs (EXP-HYDRO and Xin'an jiang model) performances in different basins using ERA5-Land and CN05.1 precipitation data.** (a–b) Spatial distribution of basins where EXP-HYDRO outperforms (yellow) or underperforms (blue) the Xin'an jiang model under ERA5-Land (a) and CN05.1 (b) forcing. (c) Sankey diagram showing the consistency of best-performing PBMs across datasets. (d) Number of basins where EXP-HYDRO performs better or worse than Xin'anjiang model in each of the 9 major river basins under both forcings. (e)

Number of basins where EXP-HYDRO performs better or worse than Xin'an jiang model in each of the 7 climate sub-regions under both forcings.

…"

**Comment 22:** *Line 467: These two references do not support he statement that NSE>=0.55 is good. Knoben says 0.50, Newman says it means the model has some skill, and NSE=0.8 shows reasonably good performance. Please clarify.*

**Response:**

Thank you for your correction. We agree that our previous interpretation of the cited studies was too strong. As clarified in Newman et al. (2015), an NSE of 0.55 is considered to indicate some model skill, whereas NSE = 0.8 reflects reasonably good performance. We have revised the manuscript to more accurately reflect this distinction. The sentence has been modified to:

"…

Specifically, when using ERA5-Land precipitation data, the median NSE of LSTM across 544 basins reaches 0.57, which suggests that the model demonstrates some skill in most basins, according to the evaluation threshold proposed by Newman et al. (2015).

…"

**Comment 23:** *Line 534-537: Indeed, since XAJ does not have snow process representation?*

**Response:**

Thank you for your comment. Indeed, the Xin'an jiang (XAJ) model does not explicitly represent snow accumulation and melt processes, which likely contributes to the inferior performance of the XAJ+LSTM hybrid scheme in snow-affected basins. We agree that this limitation should be acknowledged more clearly in the manuscript. The relevant contents after modification are as follows:

"…

This distinction highlights a key difference in the suitability of hybrid model structures across basins with varying snow dynamics. Although both EXP and XAJ are widely used conceptual hydrological models, they differ significantly in terms of process representation. The EXP-HYDRO model includes a simplified snow accumulation and melt component, whereas the Xin'an jiang (XAJ)

model, in its original and simplified forms, does not explicitly account for snow-related processes. As a result, in snow-affected basins, particularly those basins located in high-altitude regions with seasonal snowpack, the hybrid scheme coupling EXP and LSTM generally achieves superior predictive performance compared to the XAJ-LSTM hybrid. This performance gain can be attributed to the EXP model's capacity to represent key aspects of snow dynamics, enabling the hybrid model to capture runoff behavior more realistically in these regions. These findings underscore the importance of selecting appropriate process components within hybrid models to align with the dominant hydrological processes of a region, such as snow accumulation and melt in high altitude mountainous areas.

…"

**Comment 24:** *Figure 11: there are two B panels. 2nd B panel missing numbers of the overall source/destination bins. Same comments as for Figure 8.*

**Response:**

Thank you for your correction. We have corrected the subfigure labels to avoid duplication. The Sankey diagram previously labeled as Figure 11(b) has now been relabeled as Figure 11(c), and the subsequent subplots have been updated accordingly. The modified Figure 11 and its caption are as follows:

"….

[Figure]

**Figure 11. Comparison of hybrid model performances across basins using ERA5-Land and CN05.1 precipitation data**. (a) Spatial distribution of the best-performing hybrid models under ERA5-Land forcing. (b) Spatial distribution under CN05.1 forcing. (c) Sankey diagram showing the consistency of best-performing models across datasets. (d) Number of best-performing basins for each model in the 9 major river basins. (e) Number of best-performing basins in the 7 climate sub- regions.

…."

We would like to thank the editors and reviewers once again for their valuable suggestions on our manuscript. We have incorporated these suggestions into the revised manuscript. Looking forward to hearing from you.

Chunxiao Zhang

Corresponding author

E-mail address: zcx@cugb.edu.cn

References:

Arsenault, R., Martel, J.-L., Brunet, F., Brissette, F., and Mai, J., 2023. Continuous streamflow prediction in ungauged basins: long short-term memory neural networks clearly outperform traditional hydrological models, Hydrol. Earth Syst. Sci., 27, 139–157, https://doi.org/10.5194/hess-27-139-2023.

Jiang, S., Zheng, Y., Solomatine, D., 2020. Improving AI System Awareness of Geoscience Knowledge: Symbiotic Integration of Physical Approaches and Deep Learning. Geophysical Research Letters 47, e2020GL088229. https://doi.org/10.1029/2020GL088229

Kingma, D.P., Ba, J., 2014. Adam: A method for stochastic optimization. arXiv Preprint arXiv:1412.6980.

Kratzert, F., Klotz, D., Herrnegger, M., Sampson, A.K., Hochreiter, S., Nearing, G.S., 2019. Toward Improved Predictions in Ungauged Basins: Exploiting the Power of Machine Learning. Water Resources Research 55, 11344–11354. https://doi.org/10.1029/2019WR026065

Kratzert, F., Klotz, D., Shalev, G., Klambauer, G., Hochreiter, S., and Nearing, G., 2019. Towards learning universal, regional, and local hydrological behaviors via machine learning applied to large-sample datasets, Hydrol. Earth Syst. Sci., 23, 5089–5110, https://doi.org/10.5194/hess-23-5089-2019.

Miao Y , Wang A .Evaluation of Routed-Runoff from Land Surface Models and Reanalyses Using Observed Streamflow in Chinese River Basins.Journal of Meteorological Research, 2020, 34(1):73-87. https://10.1007/s13351-020-9120-z.

Newman, A.J., Clark, M.P., Sampson, K., Wood, A., Hay, L.E., Bock, A., Viger, R.J., Blodgett, D., Brekke, L., Arnold, J.R., Hopson, T., Duan, Q., 2015. Development of a large-sample watershed-scale hydrometeorological data set for the contiguous USA: data set characteristics and assessment of regional variability in hydrologic model performance. Hydrol. Earth Syst. Sci. 19, 209–223. https://doi.org/10.5194/hess-19-209-2015

Zhan, Y. , Guo, Z. , Yan, B. , Chen, K. , Chang, Z. , & Babovic, V., 2024. Physics-informed identification of pdes with lasso regression, examples of groundwater-related equations. Journal of Hydrology, 638. https://doi.org/10.1016/j.jhydrol.2024.131504

---

## Author Comment (AC6)

**Author comments on RC4- egusphere-2025-1161**

***Comment 1:*** *The question of what model structures (including those that incorporate LSTM functionality) can best represent the relevant hydrologic processes across a continental scale domain with varying landscape and hydroclimatic characteristics is an important one. Providing useful answers to this sort of question relies on some interrogation and analysis of which model structures are associated with better performance. The authors compile a dataset of basin characteristics but make no connection between that and model performance variation. This manuscript would be improved with more explanation and interpretation of how known differences in model structures relate to better or worse model performance - and better address the stated goal of providing "scientific guidance for selection and application of hydrologic models" [line 166-167]*

**Response:**

Thank you for this constructive and insightful comment. We fully agree that understanding how model structures interact with basin-specific characteristics is crucial for advancing hydrological modeling and providing meaningful guidance for model selection. In this study, although we did not classify basins based on individual catchment attributes, we performed our analysis by dividing the 544 basins according to China's nine major river systems and six climatic zones. These regional divisions already reflect important differences in landscape and hydroclimatic conditions and offer spatially coherent insights for model comparison across diverse hydrological regimes.

The main reasons we did not further stratify basins using specific catchment attributes (slope, soil type, land cover) are as follows: (1) the current attribute data are relatively limited in terms of quantity and resolution, and (2) there is no widely accepted or standardized method for catchment classification based on multivariate attributes (Jehn et al., 2020; He et al., 2023; Yang et al., 2023). Moreover, attribute-based classification may lead to spatial discontinuities, which would reduce its usefulness for practitioners aiming to apply modeling strategies in specific regions.

In further research, we plan to expand the Chinese large-sample watershed dataset by incorporating additional static watershed attributes and developing classification methods based on these attributes. This expansion aims to refine our model selection framework and evaluate the effectiveness of various watershed classification schemes in improving runoff simulation accuracy. A discussion of

these future directions has been included in the revised manuscript to highlight the potential for further enhancing the robustness and generalizability of our approach.

We have added discussion on this point in the revised manuscript, which reads as follows:

"…

Although this study does not explicitly group catchments based on physiographic or land surface attributes, the analysis framework incorporates regional differentiation by organizing the basins according to the nine major river systems and six major climatic zones in China. These divisions reflect inherent hydroclimatic and geographic diversity, and enable a meaningful comparison of model performance under varied environmental conditions.

Nonetheless, exploring how model performance varies with catchment characteristics remains an important direction for future research. With the improvement of catchment attribute datasets and the development of robust clustering methods, future studies can expand on the current work by examining how model structure suitability depends on physiographic, climatic, or hydrological conditions. This line of inquiry may provide more refined guidance for regional model selection and enhance the interpretability and generalizability of hybrid modeling approaches.

…"

**Comment 2:** *The performance of models appears to be evaluated by comparing to the results of a different (VIC) model. The practical need for this approach (lack of consistent and comprehensive streamflow data) is understandable, but a rationale that justifies how this provides meaningful insight is not provided. For example - does the analysis reflect which model structures are most similar to VIC, or do they provide some broader insight about a "true" or "best" hydrologic model for different watersheds and regions? Additional explanation and justification is needed to clarify this component of the study.*

**Response:**

Thank you for this insightful comment. We fully agree that ideally, model evaluation should be based on observed streamflow data. However, due to the scarcity and inconsistency of long-term, high-resolution observed runoff data across all 544 basins in China, which are particularly pronounced in remote or ungauged basins, it is not currently feasible to construct a comprehensive nationwide benchmark completely based on observations. In light of this limitation, we adopted the

VIC-CN05.1 runoff dataset as a reference product, considering both its spatial-temporal continuity and its demonstrated reliability in previous hydrological research in China (Miao et al., 2020; Ma et al., 2024; Wang et al., 2024; Yu et al., 2025).

To ensure that the VIC-CN05.1 runoff can reasonably represent actual streamflow dynamics, we conducted an independent verification in the Supplementary Material (Figure S2), where 15 representative basins with available observed daily runoff data (from January 1, 2015 to December 31, 2015) were compared to the corresponding VIC-CN05.1 simulated runoff series. The Pearson correlation coefficients between the VIC runoff and observed data exceeded 0.8 in nearly all basins, indicating strong agreement in temporal variability. This supports the notion that the VIC-CN05.1 product can be used as a proxy benchmark for comparative model evaluation, especially in large-sample settings where observational coverage is limited.

It is important to emphasize that the goal of our study is not to assess the fidelity of models in emulating VIC per se, but rather to understand the relative predictive skill, generalization capacity, and physical consistency of different modeling paradigms (process-based, data-driven, and hybrid) under a unified reference runoff product. While we acknowledge the limitations inherent in using simulated runoff, this framework enables a fair, consistent, and spatially extensive performance comparison across models.

To address your concern and avoid possible misunderstanding, we have clarified this rationale in the revised manuscript and explicitly stated that using simulated runoff introduces a trade-off between spatial completeness and observational accuracy. We have also acknowledged that future studies should aim to validate the models further using observed streamflow datasets as more consistent data become available. The details of our revised content are as follows:

"…

About Data: used to clarify the sources and characteristics of runoff data

The streamflow data used for model training and evaluation in this study are derived from the VIC-CN05.1 runoff product, which was generated by the VIC hydrological model driven by CN05.1 precipitation. It should be noted that this is not observational streamflow data, but a simulated product that serves as a consistent and spatially complete surrogate in the absence of publicly available observed daily streamflow records for a large number of catchments in China. While VIC-CN05.1 has been evaluated in previous studies, its use as a reference introduces potential structural

bias, particularly when comparing models forced by the same precipitation product.

About Discussion: add discussion corresponding to runoff data

One limitation of this study lies in the use of a simulated runoff product (VIC-CN05.1), generated using the CN05.1 precipitation data, as the reference for model evaluation. This introduces a structural bias that may favor models driven by CN05.1 due to internal consistency between inputs and targets. As a result, the observed performance advantage of CN05.1-driven models may not necessarily indicate the intrinsic quality of the precipitation product. The meteorological forcing comparison in this study should therefore be interpreted as an exploration of input-output consistency effects rather than a definitive evaluation of forcing product accuracy. Future studies incorporating additional meteorological datasets and observed streamflow records will be important for validating and extending these findings.

…"

*Comment 3:* *It is not entirely clear what the value of using the ERA5 and CN05.1 forcing datasets is when the evidence suggests the CN05.1 dataset provides better consistency with local conditions and better performance (albeit in comparison to a model also forced with CNO5.1). It seems that if the performance were being evaluated against true runoff or streamflow observations, model performance (such as with NSE as in Figure 7) would provide a meaningful basis for interpretation. As presented, the inclusion of models driven by ERA5 forcing complicates (and potentially obscures) a clear interpretation of the appropriateness of model structures. For example - how does training/calibrating a model that uses the ERA5 forcing against a target based on the CN05.1 forcing generate reliable information?*

**Response:**

Thank you for this insightful comment. We acknowledge that using ERA5-Land forcing data as input while using runoff outputs from the VIC-CN05.1 product as target may lead to a mismatch in the input-target consistency.

The primary motivation for including both CN05.1 and ERA5-Land precipitation as model inputs was not to determine which meteorological product is superior in an absolute sense, but rather to investigate the sensitivity and robustness of different modeling approaches (process-based, deep

learning, and hybrid) to variations in forcing data. This approach allows us to assess how input data quality and consistency influence model performance under otherwise identical settings. It also highlights the degree to which models can generalize across datasets of different spatial and temporal characteristics. Indeed, CN05.1-based models are expected to perform better when compared against VIC-CN05.1 runoff, since both originate from the same meteorological forcing. However, this does not invalidate the value of ERA5-Land -driven experiments. In fact, these models help us understand how well each modeling paradigm can accommodate inconsistencies or mismatches between input data and target outputs, which is highly relevant in practical applications, especially in data-sparse regions or when switching between data sources is necessary.

To avoid potential misinterpretation, we have revised the relevant statements in the Results section to avoid implying that CN05.1 is "better" than ERA5-Land. Instead, we now emphasize that CN05.1 yields better consistency with the target runoff used in this study (VIC-CN05.1), and that ERA5-Land experiments primarily serve to test model robustness and adaptability. We have also added the following clarification in the revised manuscript:

"…

It should be noted that while CN05.1-based models naturally align more closely with the VIC-CN05.1 runoff product used as reference, the inclusion of ERA5-Land forcing data in this study is intended to evaluate the sensitivity of model performance to meteorological input differences. This helps highlight the robustness and generalization capabilities of different modeling paradigms when faced with inconsistent or lower-fidelity inputs, conditions that are often encountered in practice, especially in ungauged or data-sparse basins. Therefore, the comparison between CN05.1 and ERA5-Land inputs should be interpreted as an assessment of model adaptability, rather than an absolute evaluation of forcing quality.

…"

***Comment 4:*** *The water budget analysis is a valuable complement to the runoff performance comparisons. However, it is unclear exactly how the closure error should be interpreted. The water balance presented implies the closure error may include changes in watershed storage (groundwater, snow, deep soil, etc) that may or may not be well represented in the models. Some further*

*clarification and explanation that justifies the interpretation that the "smaller value of epsilont, the*

*better the water budget balance of the basin" [lines 580-581]*

**Response:**

Thank you for the reminder. We agree that the presented water balance formulation, expressed as $\varepsilon = |P - ET - Q|$, simplifies the full hydrological water balance by not explicitly including the change in storage ($\Delta S$), which may encompass components such as soil moisture, groundwater, snowpack, or deep aquifer storage. In our study, this simplified formulation was intended as a diagnostic metric to evaluate the apparent water budget imbalance using the available data products, under the assumption that storage changes over longer periods are comparatively small or tend to average out. However, we acknowledge that this assumption may not hold for shorter time windows or in basins with strong seasonal storage dynamics (snow-affected basins, deep soils, or those influenced by significant groundwater).

To avoid confusion and over-interpretation of the $\varepsilon$, we have revised the manuscript to clarify its limitations and now explicitly state that $\varepsilon$ captures both the imbalance in modeled fluxes and the effects of neglected or unmodeled storage changes. Therefore, while a smaller ε suggests a more internally consistent model in terms of the fluxes represented, it does not imply perfect mass closure in a strict hydrological sense. The relevant sentence in the manuscript has been updated as follows: "…

It should be noted that the ε used in this study does not account for changes in water storage ($\Delta S$) such as snow accumulation, soil moisture, or groundwater levels. Thus, $\varepsilon$ represents a combination of actual water balance error and the effect of unmodeled or unobserved storage variations. A smaller ε value generally indicates a more internally consistent representation of fluxes ($P$, $ET$, and $Q$), but it should not be interpreted as strict mass balance closure.

…

**Comment 5:** *In general the figures are well done and informative. Their effectiveness could be improved in many cases by 1) enlarging the axis and label text and 2) providing more informative and more complete captions and annotations. In many figures (e.g. Figure 11) it is difficult directly discern the many types of information being presented.*

**Response:**

Thank you for your constructive suggestion regarding the clarity and readability of the figures. We appreciate your positive comments about their overall quality and agree that improvements in text size and caption detail would enhance their interpretability.

In the revised manuscript, we have carefully reviewed all figures and made the following modifications as per your recommendation:

Axis and label text sizes have been enlarged uniformly across all figures to ensure clarity and readability, particularly when viewed in print.

Figure captions have been rewritten or expanded to provide more detailed descriptions of what each panel represents, the data sources used, and how to interpret key patterns or comparisons

For figures with multiple subplots (Figure 11), we have:

Corrected any sub-figure labeling issues (duplicate (b) labels). Added explicit explanations in the captions about what each subplot shows. Included improved legends or annotations within the figures where appropriate to aid interpretation.

The revised Figure 11 and its caption are as follows:

"….

[Figure]

**Figure 11. Comparison of hybrid model performances across basins using ERA5-Land and CN05.1 precipitation data**. (a) Spatial distribution of the best-performing hybrid models under ERA5-Land forcing. (b) Spatial distribution under CN05.1 forcing. (c) Sankey diagram showing the consistency of best-performing models across datasets. (d) Number of best-performing basins for each model in the 9 major river basins. (e) Number of best-performing basins in the 7 climate sub- regions.

…."

**Comment 6:** *The methods and interpretation used for the "prediction in ungaged basins (PUB)" portion of the analysis is a bit confusing. Some more specific explanation that covers 1) the intent of this analysis and 2) how it is different from the other performance comparisons would make the paper much more effective.*

**Response:**

Thank you for the reminder. To improve the clarity and effectiveness of the manuscript, we have revised the relevant sections to more clearly explain both the motivation behind the PUB test and how it differs from the other modeling strategies. The PUB test in this study is designed to evaluate the extrapolation capability of various models in situations where no observed runoff data are available for the target basin, which is an issue frequently encountered in practical hydrological modeling (Hrachowitz et al., 2013; Sivapalan et al., 2003). This setup reflects a real-world scenario in which models must rely entirely on information learned from other basins, without any local calibration, to make predictions in ungauged regions.

To simulate this condition, we employed a leave-one-cluster-out strategy. In this strategy, each cluster is randomly grouped. During the training phase, one cluster is withheld and used solely for testing. This ensures that the model's performance in these basins reflects its generalization ability rather than its fit to seen data. In contrast, the regional modeling experiment includes all basins during training and measures performance within a cross-validation framework, while the basin-specific experiment calibrates and tests models individually at each site.

We have now incorporated these clarifications into the Methods section of the revised manuscript to emphasize the unique purpose and implications of the PUB analysis. The revised specific content is as follows:

"….

Furthermore, to further assess the generalization performance of the models, a five-fold cross-validation method is implemented. Specifically, the 544 basins are divided into five relatively even clusters (with each cluster containing either 109 or 108 basins, as shown in Figure 2). The validation process is as follows: the model is trained using the training period data from the basins in four of the clusters, and its performance is validated on the test period data from the remaining cluster. This

operation is repeated in a loop, with each iteration designating a different cluster as the ungauged basin, thereby allowing for the evaluation of the predictive performance of each basin treated as an ungauged basin. This experiment design effectively simulates a Prediction in Ungauged Basins (PUB) scenario, which is a key research topic in large-sample hydrology. In this setting, no runoff observations from the target basins are used during model training, ensuring a strict spatial independence between training and testing data. The purpose of this analysis is to assess the models' ability to generalize hydrological knowledge from gauged to ungauged basins, which is critical for practical applications in data-scarce regions. Compared with regional modeling experiments, the PUB test can evaluate model robustness and transferability. By averaging the performance across five folds, this strategy offers a reliable estimate of model's generalization capability.

…."

**Comment 7:** *Lines 96-98: "type of model excels in data collaboration….This type of model performs well in data-driven collaboration..."*

**Response:**

Thank you for your reminder. The two sentences convey overlapping ideas and lead to redundancy. In the revised manuscript, we have removed the repetitive expression for clarity and conciseness. The revised sentence is as follows:

"….

This type of model excels in capturing complex patterns from large-scale data and is particularly suitable for hydrological modeling in data-rich environments (Fang et al., 2022; Kratzert et al., 2019a; Tsai et al., 2021).

…."

**Comment 8:** *Lines 127-128: "..neuralizes the process-based model and adjusts model parameters by back propagating gradients based on daily prediction results.."*

**Response:**

Thank you for your reminder. The original manuscript has lacked clarity and could be confusing to readers unfamiliar with the hybrid modeling framework. In the revised manuscript, we have rephrased the sentence to provide a more precise and comprehensible explanation of how the

differentiable hybrid model integrates neural networks with process-based model components using gradient-based optimization. The revised sentence is as follows:

"….

Specifically, this hybrid modeling approach transforms the process-based model into a differentiable form by embedding it within a neural network framework. This allows the model parameters to be optimized through gradient backpropagation using daily runoff prediction errors.

…."

**Comment 9:** *Lines 321-322: "With the continuous advancement of deep learning technology, its applications in the field of hydrology are also expanding."*

**Response:**

Thanks for your comment. The original manuscript was not clear enough and failed to provide specific insights. We have revised the sentence in the manuscript to better reflect the relevance and current significance of deep learning applications in hydrology, especially in the context of large-sample hydrological modeling and hybrid model development. The revised sentence is as follows:

"….

Deep learning techniques have recently gained significant attention in hydrological modeling due to their ability to learn complex, nonlinear relationships directly from data without relying on explicit physical assumptions. In this study, the long short-term memory (LSTM) network is adopted as a representative purely data-driven model.

…."

We would like to thank the editors and reviewers once again for their valuable suggestions on our manuscript. We have incorporated these suggestions into the revised manuscript. Looking forward to hearing from you.

Chunxiao Zhang

Corresponding author

E-mail address: zcx@cugb.edu.cn

References:

Fang, K., Kifer, D., Lawson, K., Feng, D., Shen, C., 2022. The Data Synergy Effects of Time Series Deep Learning Models in Hydrology. Water Resour. Res. 58, e2021WR029583. https://doi.org/10.1029/2021WR029583

He, Z., Shook, K., Spence, C., Pomeroy, J. W., and Whitfield, C., 2023. Modelling the regional sensitivity of snowmelt, soil moisture, and streamflow generation to climate over the Canadian Prairies using a basin classification approach, Hydrol. Earth Syst. Sci., 27, 3525–3546, https://doi.org/10.5194/hess-27-3525-2023.

Hrachowitz, M., Savenije, H., Blöschl, G, McDonnell, J., Sivapalan, M., Pomeroy, J., et al. (2013). A decade of Predictions in Ungauged Basins (PUB)—A review. Hydrological sciences journal, 58(6), 1198–1255.

Jehn, F. U., Bestian, K., Breuer, L., Kraft, P., and Houska, T., 2020. Using hydrological and climatic catchment clusters to explore drivers of catchment behavior, Hydrol. Earth Syst. Sci., 24, 1081–1100, https://doi.org/10.5194/hess-24-1081-2020.

Kratzert, F., Klotz, D., Herrnegger, M., Sampson, A.K., Hochreiter, S., Nearing, G.S., 2019. Toward Improved Predictions in Ungauged Basins: Exploiting the Power of Machine Learning. Water Resources Research 55, 11344–11354. https://doi.org/10.1029/2019WR026065

Ma X , Wang A ., 2024. Evaluation and Uncertainty Analysis of the Land Surface Hydrology in LS3MIP Models Over China. Earth & Space Science, 11(7). https://10.1029/2023EA003391.

Miao Y , Wang A ., 2020. Evaluation of Routed-Runoff from Land Surface Models and Reanalyses Using Observed Streamflow in Chinese River Basins.Journal of Meteorological Research, 34(1):73-87. https://10.1007/s13351-020-9120-z.

Sivapalan, M., Takeuchi, K., Franks, S., Gupta, V., Karambiri, H., Lakshmi, V., et al. (2003). IAHS decade on Predictions in Ungauged Basins (PUB), 2003–2012: Shaping an exciting future for the hydrological sciences. Hydrological sciences journal, 48(6), 857–880.

Tsai, W.-P., Feng, D., Pan, M., Beck, H., Lawson, K., Yang, Y., Liu, J., Shen, C., 2021. From calibration to parameter learning: Harnessing the scaling effects of big data in geoscientific modeling. Nat. Commun. 12, 5988. https://doi.org/10.1038/s41467-021-26107-z

Wang Z , Li M , Zhang X., 2024. Prediction of long-term future runoff under multi-source data assessment in a typical basin of the Yangtze River.Journal of Hydrology: Regional Studies,

56(000). https://10.1016/j.ejrh.2024.102053.

Yang M , Olivera F. , 2023. Classification of watersheds in the conterminous United States using shape-based time-series clustering and Random Forests. Journal of Hydrology, 620(Pt.A). https://10.1016/j.jhydrol.2023.129409.

Yu X , Zhang Q , Zeng X., 2025. The distribution and driving climatic factors of agricultural drought in China: Past and future perspectives.Journal of Environmental Management, 377. https://10.1016/j.jenvman.2025.124599.